# OSERVE: Accelerating LLM Serving via Spatial-Temporal Workload Orchestration

Youhe Jiang [* 1]  Fangcheng Fu [* 2]  Taiyi Wang [* 1]  Guoliang He [1]  Eiko Yoneki [1]

## Abstract

Serving Large Language Models (LLMs) can benefit immensely from parallelizing both the model and input requests across multiple devices, but incoming workloads exhibit substantial *spatial* and *temporal* heterogeneity. Spatially, workloads comprise heterogeneous requests with varying compute and memory demands. Temporally, workload composition varies over time. Nevertheless, existing systems typically assume spatially uniform and temporally stable workloads, employing a homogeneous, static model deployment. This mismatch between the assumption and real-world spatial-temporal heterogeneity results in suboptimal performance. We present OSERVE, an LLM serving system with *heterogeneous* and *flexible* model deployment that addresses both spatial and temporal heterogeneity. First, OSERVE introduces a novel *workload-aware scheduling algorithm* that optimizes heterogeneous model deployments according to real-time workload characteristics. Second, OSERVE proposes an efficient *workload-adaptive switching method* that migrates model deployments in response to predicted workload changes. Experiments on real-world traces show that OSERVE improves performance by up to $2\times$ (average: $1.5\times$) compared to state-of-the-art serving systems.

## 1. Introduction

Large Language Models (LLMs) such as OPT (Zhang et al., 2022), Llama (Dubey et al., 2024), gpt-oss (Agarwal et al., 2025), Gemini (Reid et al., 2024), Claude (Anthropic, 2024), and Mixtral (Jiang et al., 2024) have demonstrated excep-

[*]Equal contribution [1]Department of Computer Science, University of Cambridge, Cambridgeshire, UK [2]School of Artificial Intelligence, Shanghai Jiao Tong University, Shanghai, China. Correspondence to: Eiko Yoneki <eiko.yoneki@cl.cam.ac.uk>.

*Proceedings of the 43rd International Conference on Machine Learning*, Seoul, South Korea. PMLR 306, 2026. Copyright 2026 by the author(s).

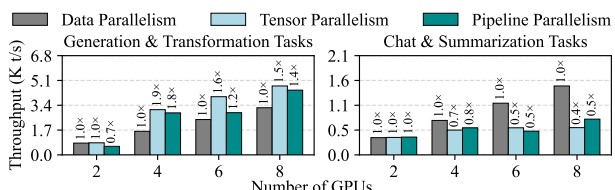

*Figure 1.* Performance comparisons of different parallelism strategies across resource allocations and workload types. The two workload types are subsampled from real-world traces in the Azure Public Dataset (Patel et al., 2024).

tional performance across a range of advanced applications (Peng et al., 2023; Jeon & Lee, 2023; GitHub, 2024). To democratize LLMs, it has become a timely and important topic to optimize the efficiency of LLM serving.

With LLMs deployed across increasingly diverse applications, inference workloads exhibit substantial heterogeneity along two key dimensions: (**i**) **Spatial heterogeneity:** Workloads comprise heterogeneous requests with significant variance in resource demands across concurrent requests—certain requests are compute-intensive (e.g., chat and summarization tasks with short output lengths), while others are memory-intensive (e.g., generation and transformation tasks with long output lengths) (Bai et al., 2025; Gao et al., 2024; Zhao et al., 2024c; Agrawal et al., 2024b; Jiang et al., 2025a). (**ii**) **Temporal heterogeneity:** Request composition and arrival rates fluctuate dynamically over time in response to evolving user behavior and application usage patterns (Patel et al., 2024; Jaiswal et al., 2025b).

To meet the substantial computational and memory requirements, LLMs are commonly deployed in a distributed manner using parallelism strategies (Li et al., 2024; Miao et al., 2025), including data parallelism (model replication) (Li et al., 2023), tensor parallelism (Shoeybi et al., 2019), and pipeline parallelism (Huang et al., 2019). However, these strategies exhibit distinct trade-offs when serving workloads with spatial and temporal heterogeneity.

**Spatial trade-offs.** As illustrated in Figure 1, the optimal model deployment configuration (i.e., resource allocation and parallelism strategy) varies significantly across workload types. Compute-intensive workloads with short outputs benefit from data parallelism to maximize computational throughput, whereas memory-intensive workloads with long

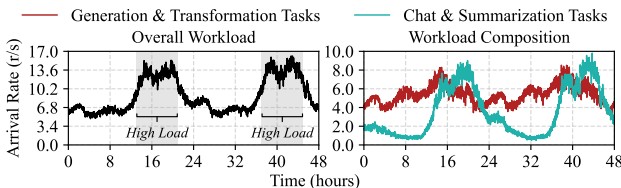

*Figure 2.* Temporal evolution of workload composition and arrival rates derived from real-world traces in the Azure Public Dataset (Patel et al., 2024).

outputs favor model parallelism to distribute memory requirements. Existing systems that employ **homogeneous** deployment configurations (Yu et al., 2022; Sun et al., 2024) across all model replicas fail to account for the diverse parallelism and resource demands from different workloads, inevitably sacrificing efficiency.

**Temporal trade-offs.** As illustrated in Figure 2, workload composition and arrival rates exhibit significant temporal variations. For instance, tasks with short outputs may predominate during business hours with high arrival rates, while tasks with long outputs increase in the evening with lower traffic, leading to shifting resource demands across different periods (Wang et al., 2024a; Stojkovic et al., 2024). Existing systems that employ **static** deployment configurations (Kwon et al., 2023; Li et al., 2023) fail to accommodate these time-varying demands in resource allocation and parallelism strategies, resulting in suboptimal resource utilization and performance degradation.

Motivated by these, this work develops OSERVE, an efficient LLM serving system that tackles the workload heterogeneity on both the **spatial** and **temporal** dimensions. Essentially, OSERVE makes two innovations as follows.

**I1: Workload-aware scheduling to address spatial heterogeneity with heterogeneous model deployment.** To handle the mixed workload composition, OSERVE deploys model replicas with non-unique resources and strategies, offering the possibility to distinguish the serving of different workload types spatially. However, this heterogeneous approach adds complexity in determining the optimal model deployment, and needs to meticulously assign the incoming requests across the heterogeneous replicas for workload balance. To achieve so, we formulate the search for heterogeneous model deployments as a *constrained optimization problem* and propose a *two-level workload-aware scheduling* algorithm. This approach co-optimizes model deployment with workload assignment, ensures that each model replica is configured and utilized in the best possible way to meet the specific demands of different workloads.

**I2: Workload-adaptive switching to address temporal heterogeneity with flexible model deployment.** In response to the dynamicity in workload composition and arrival rates, OSERVE enables adaptively switching the model deployment (as well as the request assignment) to gain flex-

ibility on the temporal dimension. Two major efforts are made to do this. Firstly, we devise a *fine-grained time-series forecasting*, which accurately predicts how the workloads change in the next time interval. Secondly, to expedite the switching of model deployment, we introduce a *ad hoc model switching* method, which avoids re-loading the huge model from scratch, but leverages the faster GPU-GPU network connections to transfer model parameters.

We conduct experiments to compare OSERVE with vLLM, Llumnix, and Dynamo on different real-world traces using popular LLMs with up to 70B parameters. Empirical results show that OSERVE reduces the end-to-end P99 tail latency and improves system throughput by up to $2\times$ and on average $1.5\times$ compared to state-of-the-art LLM serving systems.

## 2. Background

**Workload heterogeneity.** LLMs are designed to support a diverse range of applications, and these different inference workloads exhibit heterogeneity in terms of input and output sequence lengths (Naveed et al., 2023; Hadi et al., 2024). For example, chat, information extraction, and document summarization tasks, as well as requests from Burst-GPT and MMLU, typically have short output lengths (Wang et al., 2024a; Hendrycks et al., 2020; Patel et al., 2024). Conversely, code/content generation, transformation, and reasoning tasks, along with requests from ShareGPT and WildChat, usually have long output lengths (Zheng et al., 2023; Zhao et al., 2024a; Gao et al., 2024; Jain et al., 2024).

**Phases of LLM inference.** Given an input prompt, LLM inference consists of two phases: The prefill phase processes the prompt to compute the key-value (KV) cache and generates the first token in a single step, and the decoding phase takes the last generated token and KV cache as inputs to generate subsequent tokens (Vaswani, 2017). Unlike the prefill phase, the decoding phase generates tokens step-by-step, which makes it more memory-bandwidth-bound than the compute-intensive prefill phase (Zhao et al., 2024b).

**Parallelisms.** To parallelize the model over multiple GPUs, there are three prevalent forms of parallelisms, which are data (i.e., model replication) (Li et al., 2023; Liu et al., 2024), tensor (Shoeybi et al., 2019), and pipeline parallelism (Huang et al., 2019). Different parallelisms come with certain trade-offs. Numerous studies (Li et al., 2023; Zhong et al., 2024; Jiang et al., 2023; Miao et al., 2024) have investigated how to deduce the hybrid parallelism strategy by meticulously enumerating many possible combinations.

**Job scheduling in clusters.** There is also a line of research that considers the job scheduling in clusters (Isard et al., 2009; Schwarzkopf et al., 2013; Delimitrou & Kozyrakis, 2013; 2014). However, our work focuses on the request scheduling for LLM serving, which has a different goal.

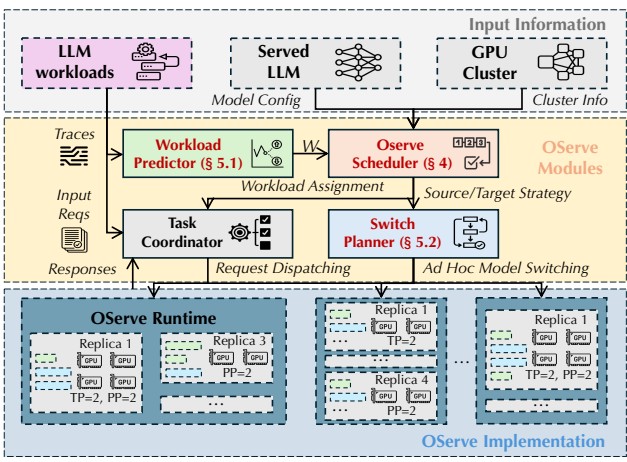

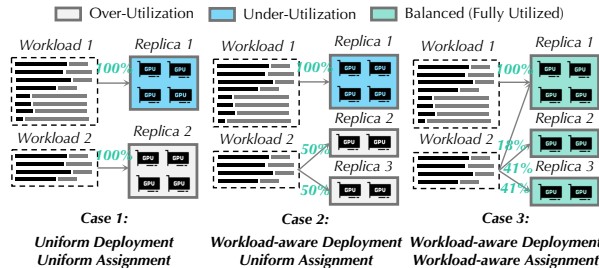

*Figure 3.* OSERVE system overview.

*Figure 4.* Example of model deployment and workload assignment.

Extended related work is provided in Appendix A.

## 3. OSERVE Overview

The architecture overview of OSERVE is shown in Figure 3. OSERVE contains three essential components: The workload predictor (§5.1), OSERVE scheduler (§4), and switch planner (§5.2). The overall routine is as follows. (**1**) **Workload prediction:** The workload predictor takes the LLM workloads' historical traces as input, based on which it differentiates between different workload types and predicts their corresponding request arrival rates in the next time span (in our case, one minute). The workload is considered stable during this time span given the short duration of each span (Duan et al., 2024). (**2**) **Strategy deduction:** The scheduler takes the cluster information, model configuration, and estimated workloads as input, formulates the flow network, and deduces the optimal serving strategy. The OSERVE engine then deploys the LLM and performs request dispatching based on the scheduling result. (**3**) **Strategy switch:** Once the workload predictor provides the workload features for the subsequent time span, the scheduler searches for the corresponding serving strategy and outputs it to the switch planner. The switch planner utilizes the source (previous) and target (current) strategies to obtain the optimal switch plan. The engine then implements model parameter switching based on the given instructions.

## 4. Workload-aware Scheduling

### 4.1. Scheduling Problem Statement

To support efficient workload-aware LLM serving, the scheduling algorithm must determine two essential allocations: (**i**) *Model deployment*, which specifies the resource allocations and parallelism strategies for multiple model replicas; and (**ii**) *workload assignment*, which determines the distribution of workload types across different model

replicas. We term a solution to these two components a *serving strategy*. As shown in Figure 4, different serving strategies yield different performance results. To tackle the complexity of this problem, we propose a two-level workload-aware scheduling algorithm:

**Lower-level optimization (§4.2):** Given a specific model deployment, the lower-level formulates the workload assignment problem as a directed flow network, and applies a max-flow algorithm to maximize serving performance.

**Upper-level optimization (§4.3):** Based on the lower-level outcome, the upper-level applies flow network-guided generation to obtain the optimal model deployment. This iteratively refines resource allocation and parallelism strategy based on system bottlenecks identified by the flow network.

Together, these two levels form a cohesive optimization loop. We illustrate the scheduling process with a simple example in Appendix C.

### 4.2. Lower-level Workload Assignment

The lower-level of our algorithm determines the optimal workload assignment for a specific model deployment.

**Flow network formulation.** We construct a flow network in which the source node $\mathcal{S}$ supplies all incoming requests and the sink node $\mathcal{T}$ represents completed requests. Each workload type $j$ corresponds to a workload node $w_j$, and each model replica $k$ is represented by two nodes, $c_k^{in}$ and $c_k^{out}$, where $n_{k,j}$ denotes the processing capacity of replica $k$ for workload type $j$. To facilitate per-workload assignments, we introduce intermediate nodes $i_{k,j}$. The flow network comprises four types of edges: (**i**) From the source $\mathcal{S}$ to each workload node $w_j$, with capacity $\lambda_j$ equal to the total number of incoming requests of type $j$; (**ii**) from each workload node $w_j$ through an intermediate node $i_{k,j}$ to $c_k^{in}$, with capacity $e_{k,j}$ representing the maximum number of type-$j$ requests assignable to replica $k$; (**iii**) from $c_k^{in}$ to $c_k^{out}$, with normalized capacity $M_k$ to accommodate mixed workloads (detailed below); and (**iv**) from $c_k^{out}$ to the sink $\mathcal{T}$, with sufficiently large capacity to ensure that all processed requests can exit the system. Following prior work (Patel et al., 2024; Jaiswal et al., 2025a; Lin et al., 2024), we perform one-time profiling (detailed in Appendix D) to estimate the node ca-

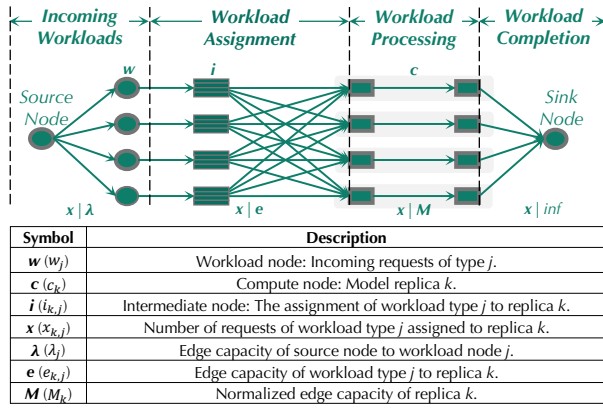

| Symbol | Description |
|---|---|
| $\boldsymbol{w}\ (w_j)$ | Workload node: Incoming requests of type $j$. |
| $\boldsymbol{c}\ (c_k)$ | Compute node: Model replica $k$. |
| $\boldsymbol{i}\ (i_{k,j})$ | Intermediate node: The assignment of workload type $j$ to replica $k$. |
| $\boldsymbol{x}\ (x_{k,j})$ | Number of requests of workload type $j$ assigned to replica $k$. |
| $\boldsymbol{\lambda}\ (\lambda_j)$ | Edge capacity of source node to workload node $j$. |
| $\boldsymbol{e}\ (e_{k,j})$ | Edge capacity of workload type $j$ to replica $k$. |
| $\boldsymbol{M}\ (M_k)$ | Normalized edge capacity of replica $k$. |

*Figure 5.* Illustration of the flow network. $a \mid b$ denotes that $a$ is the used capacity out of $b$.

pacity $n_{k,j}$ and edge capacity $e_{k,j}$ for each replica-workload pair $(k, j)$, capturing the maximum achievable throughput per workload type based on parallelism strategies. An illustration of the flow network is provided in Figure 5.

**Edge capacity normalization.** To handle mixed workloads on a single model replica, we compute the least common multiple $M_k$ of all $n_{k,j}$ for replica $k$. By setting the capacity of $(c_k^{in}, c_k^{out})$ to $M_k$, we ensure that each type-$j$ request consumes $M_k/n_{k,j}$ units of capacity. For example, if replica $k$ can process 80 type-1 requests and 50 type-2 requests per unit time, we set the capacity of $(c_k^{in}, c_k^{out})$ to the least common multiple, i.e., 400. Thus, one type-1 request consumes $400/80 = 5$ units of capacity while one type-2 request consumes $400/50 = 8$ units.

**Formulation of constraints.** We denote by $x_{k,j}$ the number of requests of workload type $j$ assigned to replica $k$, and impose the following constraints to ensure a feasible and optimal assignment:

*C1: Workload demand constraint.* This constraint ensures that the total assignment does not exceed the incoming request volume for each workload type: $\sum_{k=1}^{M} x_{k,j} \leq \lambda_j,\ \forall j$. This aligns with the supply-side limitation, preventing assignment of more type-$j$ requests than are actually available.

*C2: Edge capacity constraint.* This constraint enforces per-replica capacity limits for individual workload types: $x_{k,j} \leq e_{k,j},\ \forall k, j$. This reflects the replica-level, per-workload capacity derived from the profiling step, ensuring that the number of type-$j$ requests directed to replica $k$ does not exceed the configured edge capacity.

*C3: Node capacity sharing constraint.* This constraint governs the joint processing capacity when a replica handles multiple workload types simultaneously: $\sum_{j=1}^{J} \frac{x_{k,j} \cdot M_k}{n_{k,j}} \leq M_k,\ \forall k$, where the least common multiple $M_k$ normalizes capacities across all workloads for replica $k$. This provides a compositional capacity limit, ensuring that the combined

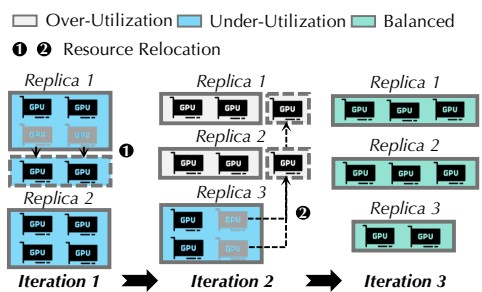

*Figure 6.* Example of flow network guided generation.

resource consumption of different workload types does not exceed the total processing capability of each replica.

Given the three constraints and the flow network formulation, we solve the resulting max-flow problem using the *preflow-push algorithm* (Cheriyan & Maheshwari, 1989), which computes the optimal workload assignment $\{x_{k,j}\}$.

### 4.3. Upper-Level Model Deployment

Based on the lower-level optimization, the upper-level problem determines the optimal model deployment to maximize overall serving performance.

**Problem formulation.** Consider a cluster of $D$ GPUs with the objective of forming $R$ model replicas (i.e., data parallelism). Let $\{d_r\}_{r=1}^{R}$ denote the number of GPUs assigned to each replica and $\{s_r\}_{r=1}^{R}$ the corresponding parallelism strategies (i.e., tensor and pipeline parallelism), subject to: $\sum_{r=1}^{R} d_r = D,\ \forall r$. For any configuration $\{d_r, s_r\}_{r=1}^{R}$, the lower-level optimization (§4.2) yields the maximum achievable throughput $\Phi(\{d_r, s_r\})$. The upper-level problem thus seeks to solve: $\max_{\{d_r, s_r\}} \Phi(\{d_r, s_r\})$.

A brute-force approach enumerating all valid configurations is feasible for small $D$ and $R$, but becomes intractable as the search space grows. To address this, we propose a flow network guided generation method that iteratively searches for the optimal deployment configuration.

**Initialization.** Prior to the search process, we initialize a uniform model deployment by allocating an identical number of GPUs to each replica and employing pure tensor parallelism. The GPU count per replica is determined by the minimum memory requirement necessary to serve the model, e.g., 140 GB for a 70B model.

**Flow network guided generation.** To efficiently navigate the large search space, we leverage insights from the lower-level max-flow analysis. Specifically, after solving the max-flow problem for a given deployment, we examine the flow distribution across different replicas to identify: (**i**) Bottleneck replicas that are fully saturated and could benefit from additional GPU resources, and (**ii**) under-utilized replicas whose assigned GPUs are not fully exploited. These insights guide a heuristic refinement process in which GPUs

are reallocated from under-utilized replicas to bottleneck replicas, iteratively improving overall throughput. Each iteration consists of three essential steps:

- Identify bottleneck and under-utilized replicas by comparing actual node flow against capacity.

- Reallocate GPUs, increasing $d_r$ for bottleneck replicas and decreasing it for under-utilized ones.

- Enumerate possible parallelization strategies $\{s_r\}$ for the adjusted configuration, selecting the one that yields the highest throughput.

Figure 6 illustrates this iterative process. By continuously refining resource allocation and parallelism strategy (i.e., $\{d_r, s_r\}$) based on flow network feedback, the upper-level optimization converges toward an efficient model deployment that maximizes system throughput.

**Termination condition.** We terminate the iterative search process when no further improvements can be made. For instance, when the maximum achievable throughput remains unchanged for 20 iterations.

## 5. Workload-adaptive Switching

### 5.1. Workload Prediction

**Challenges in workload prediction.** Three key characteristics define an LLM inference workload: (**i**) Input request sequence length, (**ii**) output request sequence length, and (**iii**) request arrival rate. However, direct prediction of these variables is impractical due to task variability and complexity. For example, in the Azure Public Dataset (Patel et al., 2024), the input sequence length ranges from 1 to 7999, and the output sequence length ranges from 1 to 5000. Furthermore, given the dynamic and unpredictable nature of user interactions, the request arrival rate can exhibit substantial fluctuations within even brief time spans (Qiao et al., 2024; Stojkovic et al., 2024; Wang et al., 2025a).

**Our approach.** Motivated by previous works (Ma et al., 2018; Duan et al., 2024) on **clustering** and **statistical**-based prediction, OSERVE employs a *fine-grained time-series forecasting* approach. Rather than forecasting the exact lengths of input and output request sequence lengths and arrival rates, we focus on workload differentiation and type-specific prediction. Our method enables OSERVE to: (**i**) Distinguish between different workload types, and (**ii**) independently predict the estimated number of requests for each type over short future time spans.

**Future workload estimation.** Although it is hard to predict the concrete future workload information, we can separate the workload into different types based on historical input and output request sequence lengths, and predict each of the workload type's the arrival rate, i.e., how many requests of each type are arrived within the next time span (one minute

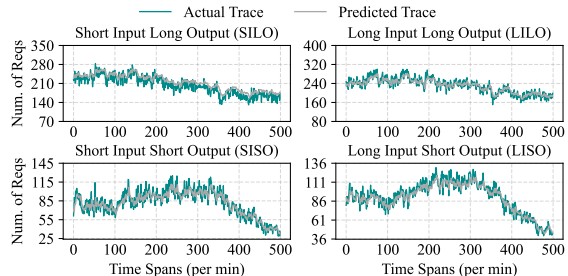

*Figure 7.* Prediction of the arrival rate (i.e., the number of requests per time span) for each workload type.

in our work[1]). Our approach consists two steps:

*S1: Process historical data.* We utilize a k-means algorithm (Ahmed et al., 2020) to categorize historical data into distinct workload types based on input and output request sequence lengths, ensuring each request is assigned to a specific workload type. We then count the number of requests that arrive within each time span for each workload type. This processed historical data enables us to establish the relationship between the total number of arrived requests and the corresponding time spans.

*S2: LSTM prediction.* We select the Long Short-Term Memory (LSTM) model (Yu et al., 2019) as our workload predictor due to its superior ability to capture long-range temporal dependencies (Chien et al., 2021; Zhang et al., 2021). It uses a sequence length of 50, enabling it to leverage data from the previous 50 minutes to predict workload patterns for the next minute. We train it using the processed historical data mentioned in the first step.

**Predictor performance.** We train an LSTM model using the two-step method with two weeks of real-world traces from the Azure Public Dataset, with 90% of the data as the training set and 10% as the test set. As shown in Figure 7, the LSTM model efficiently and accurately captures the trends of future inference workloads. The workload prediction process takes less than 30 ms with an average Relative Root Mean Square Error (RRMSE) of 5.045%, which demonstrates the effectiveness of our approach.

### 5.2. Ad Hoc Model Switching

**Challenges in model switching.** Model switching occurs when the optimal serving strategy shifts due to workload fluctuations. Due to the substantial sizes of LLMs, reloading the model from scratch is time-consuming (taking minutes as evaluated in §6). Such long switching times degrade performance under highly fluctuating workloads and diminish the benefits of adopting the new serving strategy.

**Our approach.** Since all model replicas share the same pa-

---

[1]We assume that the workload is relatively stable within each time span, which is a reasonable assumption given the short duration of each span.

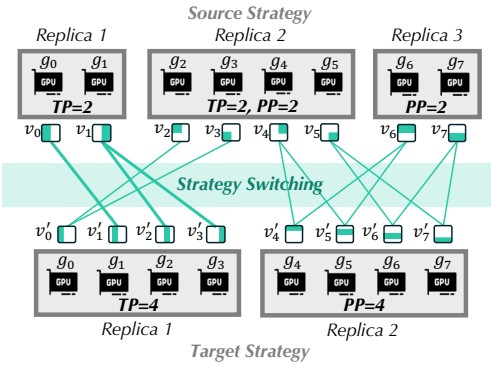

*Figure 8.* Example of strategy switching between eight GPUs ($g_0$-$g_7$), and the model parameter shards on each GPU before ($v_0$-$v_7$) and after ($v'_0$-$v'_7$) strategy switching. $v_1$ transmits part of its model parameters to $v'_2$, since it contains the model parameters $v'_2$ required; $v'_2$ can also obtain these parameters from $v_4$ and $v_5$.

rameters, OSERVE leverages high-speed GPU interconnects (e.g., NVLink and InfiniBand) to transfer model parameters between GPUs when switching deployments. We define the model deployments before and after switching as the *source strategy* and *target strategy*, and each GPU can act as both a *source device* and a *target device*. As illustrated in Figure 8, each model parameter may correspond to different numbers of source and target devices depending on the source and target strategies, resulting in numerous possible communication/switch plans. Our objective is to identify the optimal switch plan that minimizes the switching cost.

**Greedy Algorithm.** We develop a *greedy algorithm* to obtain the optimal switch plan. The algorithm initializes an empty plan for parameter transfers and sets up counters to track the volume of data communicated between each source-target device pair. For each target device, the algorithm iterates through all possible source devices, selects the one with the lowest existing communication load, updates the communication load by adding the required volume, and includes the selected source-target pair in the switch plan. This approach ensures that, for each model shard, the source device with the least data sent so far is always chosen, thereby optimizing overall communication efficiency.

The efficiency of this greedy algorithm can be enhanced using the heuristic that *intra-machine communication should always be prioritized over inter-machine communication*. NVLink (400 GB/s) handles intra-machine communication, which is typically faster than inter-machine communication using InfiniBand (IB) or RoCE (10-200 GB/s). By applying this heuristic, the iteration process is divided into two phases: First iterating among intra-machine source devices, and then, if no intra-machine sources are available, iterating among inter-machine sources. This heuristic prunes unnecessary inter-machine transmission searches for each target device, thereby accelerating algorithm convergence.

Although the heuristic-based greedy algorithm does not

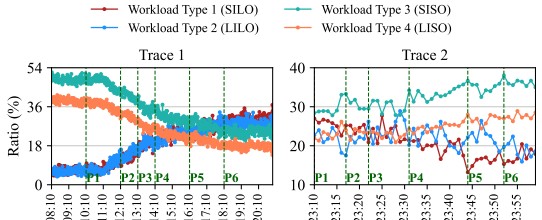

*Figure 9.* Workload type ratio changes in real traces.

guarantee an optimal switch plan in all cases, it is practical for real-world GPU clusters and effectively balances communication load across devices. The detailed procedure and pseudocode are provided in Appendix F.

**KV cache transmission.** For KV cache transmission: At switch time, (**i**) requests with short-sequence KV blocks are drained on the source deployment, whereas (**ii**) long-sequence KV blocks are migrated using the same greedy algorithm as for model parameters, leveraging fast communication links for efficient transmission. To prevent allocation stalls, we pre-allocate fixed-size KV buffers on target GPUs sized to the required KV capacity (optionally +10–20% headroom for fragmentation), and migrate KV in batched, layer-aligned chunks. Once a request drains or completes migration, the source buffers are reclaimed.

## 6. Experimental Evaluation

### 6.1. Experimental Setup

**Environments.** Our experiments are conducted on four GPU servers equipped with 8×NVIDIA H100-80GB GPUs. Within each server, the GPUs are connected via NVLink with a bandwidth of 400GB/s, and the servers are connected via InifiBand with a bandwidth of 200GB/s.

**Baselines.** To understand the system efficiency of OS-ERVE and each part of our system design, we compare it with state-of-the-art LLM serving systems: (**i**) vLLM (static) (Kwon et al., 2023): Using vLLM to serve the given LLM with a static parallel configuration. (**ii**) vLLM (reload): Enabling vLLM with ad hoc model switching (§5.2) to serve with adjusted model deployments to adapt to different inference workloads. (**iii**) Llumnix (Sun et al., 2024): Continuously rescheduling and dynamically migrating requests across instances to handle workload fluctuations. (**iv**) Dynamo+vLLM (NVIDIA, 2025): Nvidia's distributed inference framework that dynamically rebalances GPUs and routes requests/KV-cache to reduce recompute and queuing. To ensure a fair comparison, we optimize the model deployment for each baseline and report the best results.

**Models.** Similar to prior works (Zhong et al., 2024; Kwon et al., 2023), we evaluate OSERVE on LLMs with different scales, including OPT-30B and OPT-66B (Zhang et al., 2022), Llama-30B and Llama2-70B (Touvron et al., 2023).

**LLM inference workloads.** We follow previous works (Patel et al., 2024; Stojkovic et al., 2024; Zhao et al., 2024c) to generate workload traces from real-world datasets (Azure, 2024; Zhao et al., 2024a). Two traces are sampled to evaluate OSERVE, as shown in Figure 9. Trace 1 (T1) represents an 800-minute period with fluctuating workloads, while trace 2 represents a 50-minute period with different fluctuation trends. In both cases, we scale the request arrival rate based on the cluster size while maintaining the workload type ratios for each minute, ensuring the cluster capacity is neither over- nor under-utilized. And we select six time spans (P1-P6) from each trace, each with a distinct workload composition, to provide a detailed performance comparison for each specific time span.

**Evaluation metrics.** We focus on a range of percentile latencies (i.e., average, P90, and P95-99) and throughput when evaluating system performance.

## 6.2. End-to-end Performance

**End-to-end performance comparison.** Figure 10, Figure 11, and Figure 12 demonstrate the latency and throughput performance of OSERVE compared with vLLM (static) and vLLM (reload) with different configurations. OSERVE outperforms both baselines in terms of all latency and throughput metrics. In the end-to-end experiments, OSERVE improves P99 latency and throughput by up to $2.0\times$ and on average by $1.5\times$ compared to vLLM (static), and by up to $1.5\times$ and on average by $1.3\times$ compared to vLLM (reload). Across each specific time span (P1–P6) from trace 1 to trace 2, the performance gains of OSERVE over vLLM (static) and vLLM (reload) range from $1.1\times$ to $2.7\times$ and from $1.1\times$ to $1.9\times$, depending on the workload composition.

**Temporal flexibility with flexible deployment.** Compared to vLLM (static), the key advantage of OSERVE lies in its ability to predict future workloads and adjust serving strategies with minimal switching costs. OSERVE with temporal flexibility is able to modify model deployments and workload assignments in response to varying workload compositions. In comparison, a static model deployment inevitably leads to suboptimal performance during significant workload fluctuations. For example, at P1 and P6 in trace 1, the optimal model deployments for vLLM are (DP=8, TP=2)[2] and (DP=2, TP=8), as the workload at P1 benefits from greater data parallelism, while P6 benefits more from increased tensor parallelism. In this scenario, the static setup of vLLM (static) leads to performance degradation of up to $2.7\times$ compared to OSERVE. By predicting workload changes and dynamically adjusting its model deployment, OSERVE maintains optimal performance.

---

[2]Represents DP and TP degrees of 8 and 2.

**Spatial flexibility with heterogeneous deployment.** Compared to vLLM (reload), the key advantage of OSERVE lies in its ability to flexibly allocate resources, adjust parallelism strategies, and strategically assign workloads to the most suitable model replicas. OSERVE with spatial flexibility is able to implement a heterogeneous model deployment, ensuring that different workload types are directed to replicas that best match their resource needs. For example, at P5 in trace 2, vLLM (reload) uses a model deployment of (DP=4, TP=2, PP=2). In contrast, OSERVE adopts a heterogeneous model deployment, deploying four model replicas (i.e., DP=4) with different configurations: (TP=4, PP=2), (TP=2, PP=2), (TP=2, PP=1), and (TP=2, PP=1). Each workload type is routed to the most suitable replica: Types 1 and 2, benefiting from model parallelism, are directed to replicas with more resources (i.e., TP=4, PP=2), while types 3 and 4, benefiting from data parallelism, are assigned to more replicas with fewer resources (i.e., TP=2, PP=1). In this scenario, the uniform model deployment and workload assignment in vLLM (reload) result in performance degradation of up to $1.5\times$ compared to OSERVE.

**Comparison with Llumnix and Dynamo+vLLM.** We compare OSERVE with Llumnix, which integrates dynamic request migration, and Dynamo+vLLM, which autoscales prefill/decoding workers with KV-aware scheduling. As shown in Figure 17 and Figure 18 in Appendix H, when serving Llama-30B and Llama2-70B on 8–16 GPUs across multiple traces, OSERVE outperforms Llumnix by $1.32$–$1.51\times$ in P99 latency and throughput, and improves end-to-end performance over Dynamo+vLLM by 12–20%. Both baselines fail to account for the impact of model deployments (i.e., resource allocations and parallelism strategies) on LLM serving across various workload types: Llumnix does not consider different deployment configurations, while Dynamo fixes per-worker parallelism, overlooking the parallelism–workload interaction. In contrast, OSERVE co-optimizes model deployment with request scheduling, leading to significant performance improvements. Detailed experimental results and are provided in Appendix H.

**Experiments on a 32-GPU Cluster.** We further evaluate OSERVE against the baselines on a 32-GPU cluster. As shown in Figure 13, when serving the Llama2-70B model with 32 GPUs across different traces, OSERVE consistently outperforms all baselines, achieving up to a $1.9\times$ performance improvement and demonstrating strong scalability.

## 6.3. Case and Ablation Studies

**Switching cost impact on serving latency.** The switching cost can be significant with frequently fluctuating workloads. For example, in trace 2, the minimum switching interval is 5 minutes, while reloading a model takes over 50 seconds, increasing the system's average inference latency by approx-

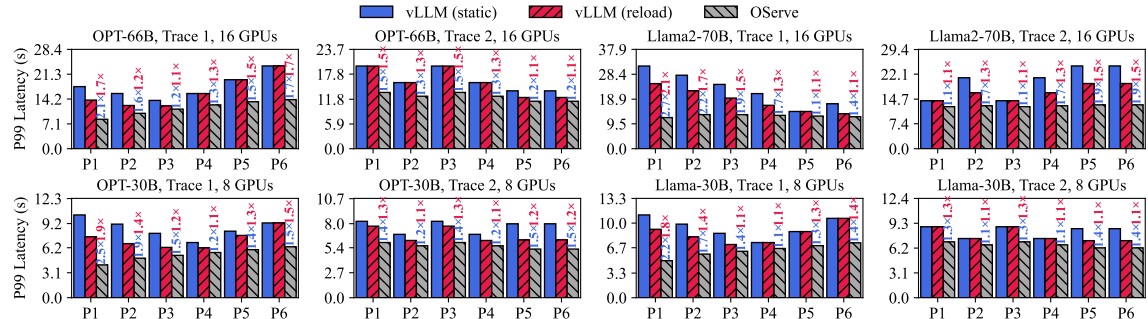

*Figure 10.* Latency results of OSERVE vs. baselines with different LLMs, GPU, and traces during different time spans (P1-P6).

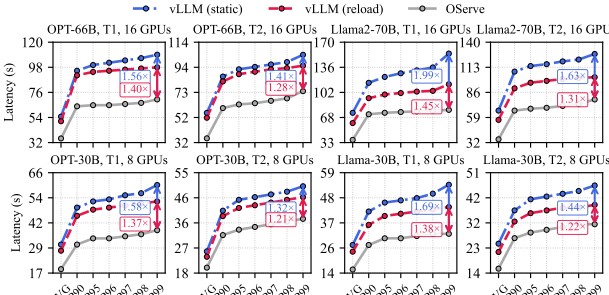

*Figure 11.* End-to-end latency results of OSERVE vs. baselines with different LLMs, GPU, and traces.

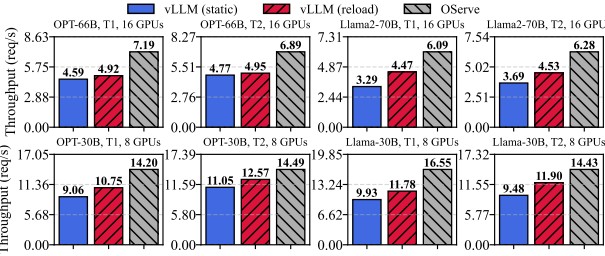

*Figure 12.* Throughput results of OSERVE vs. baselines.

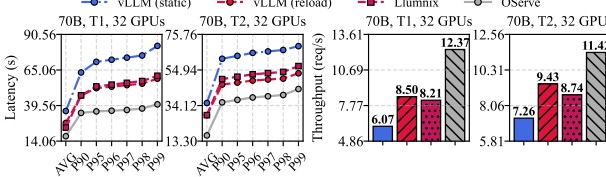

*Figure 13.* Results of OSERVE vs. baselines on 32 GPUs.

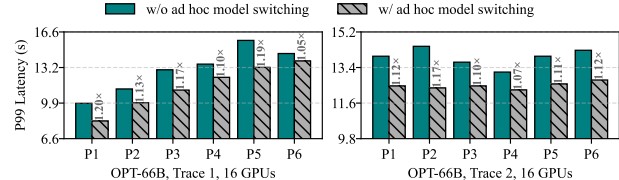

*Figure 14.* Switching cost impact on P99 latency.

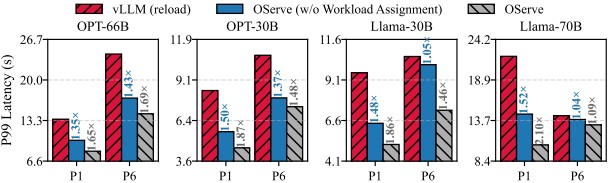

*Figure 15.* P99 latency of different models on two time spans (P1 and P6 in trace 1) with different OSERVE components.

imately 17%. However, by using the ad hoc model switching technique described in §5.2, we minimize the switching cost to around 10 seconds for any case, significantly reducing overhead and improving OSERVE's adaptability to fluctuating workloads. As shown in Figure 14, enabling ad hoc model switching reduces the system's P99 latency by up to 17% and by an average of 12% compared to naive model reloading. Note that the impact of ad hoc model switching is more significant in workloads with higher fluctuations, requiring more frequent changes in parallel configurations.

**Comparison with other workload prediction methods.** To evaluate our LSTM-based workload predictor (§5.1), we

compare it against two baselines on OPT-30B with 8 GPUs using Trace 1. The first baseline is a Moving Average (MA) predictor, which increases RRMSE to 43.375% and reduces throughput from 14.2 req/s to 10.1 req/s (41% degradation). The second baseline uses the same LSTM architecture but predicts total workload without type decomposition; this approach fails to converge during training due to high variance and unstable temporal patterns in the aggregated signal, yielding an RRMSE of approximately 40%. These results demonstrate that both simpler predictors and the absence of type decomposition fail to capture temporal dependencies and workload variations, validating our type-specific LSTM-based forecasting approach.

**Discussion on potential prediction errors.** Prediction errors are inevitable in real-world autoscaling scenarios (Pan et al., 2023). To mitigate their impact, OSERVE employs fine-grained prediction intervals (1-minute) combined with fast switching mechanisms (§5.2). Even under inaccurate predictions or unseen workload patterns, the system can quickly re-optimize deployment in the next interval, preventing long-term performance degradation.

**Ablation study.** Figure 15 shows the P99 latency of different models on two time spans (P1 and P6 in trace 1) with different OSERVE components. We start from vLLM (reload) that corresponds to optimal serving strategy of vLLM and

*Table 1.* Sensitivity to spatial heterogeneity on Llama2-70B with 16 GPUs. CV is computed over the four workload-type proportions. Speedup is measured over vLLM (static).

| Level | CV | SILO/LILO/SISO/LISO (%) | Speedup |
|---|---|---|---|
| S1 | 0.112 | 26.3/26.2/27.3/20.2 | 1.14× |
| S2 | 0.186 | 30.0/28.0/24.2/17.8 | 1.41× |
| S3 | 0.275 | 18.8/18.9/35.5/26.8 | 1.89× |
| S4 | 0.472 | 13.2/14.2/41.1/31.5 | 2.15× |
| S5 | 0.688 | 8.2/8.2/47.1/36.6 | 2.66× |

*Table 2.* Sensitivity to temporal heterogeneity on Llama2-70B with 16 GPUs. CV is the average per-type CV across consecutive time spans. Average speedup is measured over vLLM (static).

| Level | Workload Trace | CV | Avg. Speedup |
|---|---|---|---|
| T1 | S1→S2→S1 | 0.052 | 1.23× |
| T2 | S1→S3→S2 | 0.172 | 1.48× |
| T3 | S2→S4→S3 | 0.263 | 1.82× |
| T4 | S1→S4→S5 | 0.348 | 1.98× |

enables each system optimization one by one. (**i**) By enabling the heterogeneous model deployment, the P99 latency improves an average of 34% and a maximum of 52% across all cases. This heterogeneous approach allows OSERVE to adapt flexibly to different workload compositions by allocating varying resources to model replicas. For example, at P1 for OPT-66B, vLLM's optimal model deployment is (DP=4, TP=2, PP=2), while OSERVE deploys five model replicas with configurations: (TP=3, PP=2), (TP=2, PP=2), and three with (TP=2). (**ii**) Further enabling optimal workload assignment in OSERVE achieves an average improvement of 64% and a maximum of 109% in P99 latency. This optimization leverages heterogeneous model deployment by routing workloads to replicas that best match their resource requirements, thereby maximizing the utilization of the cluster's capabilities. For example, in the OPT-66B and P1 case, 100% of workloads 1 and 2 are routed to the replica with (TP=3, PP=2), 88% of workload 4 goes to the replica with (TP=2, PP=2), and the remaining 12% of workload 4 and 100% of workload 3 are handled by replicas with (TP=2).

**Sensitivity to spatial and temporal heterogeneity.** We evaluate OSERVE across a spectrum of spatial and temporal heterogeneity levels on Llama2-70B with 16 GPUs. For spatial heterogeneity, we construct five workload composition levels (S1–S5) from Azure traces with increasing skew, using the coefficient of variation (CV) of the four workload-type proportions as the indicator. Table 1 shows that OSERVE's speedup over vLLM (static) increases from 1.14× under near-uniform workloads (CV=0.112) to 2.66× under highly skewed compositions (CV=0.688). For temporal heterogeneity, we construct four traces (T1–T4) with progressively shifting workload compositions across consecutive time spans, measured by the average per-type CV. Table 2 shows that OSERVE's average speedup increases from 1.23× to 1.98× as temporal heterogeneity intensifies. These results demonstrate that workload-aware model deployment and assignment become increasingly effective as heterogeneity grows, while static baselines suffer from increasing mismatch to workload dynamics.

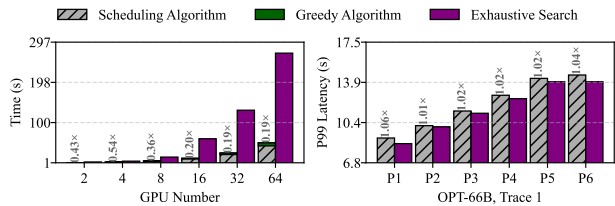

*Figure 16.* Algorithm scalability and impact on latency.

Additionally, we present a case study of the model deployments of OSERVE over different time spans in Appendix G, along with TTFT and TBT results demonstrating further improvements in key serving metrics in Appendix I. We also evaluate OSERVE on homogeneous workloads in Appendix J and discuss its extensibility to emerging parallelism strategies in Appendix K.

### 6.4. Algorithm Efficiency

**Algorithm running time.** Figure 16 shows the execution time of our scheduling algorithm (§4 scheduling + §5.2 greedy algorithm) as the number of GPUs increases. The results indicate that OSERVE scales well with the growing number of GPUs. Additionally, both algorithms are highly parallelizable, as different resource allocations and strategies are independent, allowing the execution time to decrease linearly with more CPU cores. Note that the scheduling time should be within the 1-minute prediction time span mentioned in §5.1, as these steps need to overlap to avoid delays between scheduling and prediction. If the cluster size is extremely large and the search process takes longer than one minute, the prediction time span should be adjusted, or more CPU cores should be utilized to accelerate the computation.

**Algorithm Optimality.** To evaluate the optimality of our heuristic search, we compare the scheduling results of our heuristic design in §4.3 with those from exhaustive search, treated as the *optimal* baseline. As shown in Figure 16, on a 16-GPU cluster, exhaustive search takes around 50s due to the large number of possible resource allocations and parallel strategies, while our heuristic method completes within 12s. Additionally, we benchmarked the P99 latencies of serving OPT-66B using scheduling results obtained through heuristic and exhaustive search. As shown in Figure 16, the P99 latency gap between the two methods is within 6%, demonstrating the effectiveness of our heuristic approach.

## 7. Conclusion

We propose OSERVE, an LLM serving system that adapts to workload heterogeneity through runtime rescheduling and model deployment switching. It mitigates **temporal heterogeneity** within workload dynamics via predictive runtime model deployment switching, and addresses **spatial heterogeneity** in workload distribution through heterogeneous model deployment and adaptive workload assignment. Results demonstrate that OSERVE improves performance by up to 2× (average: 1.5×) compared to existing systems.

## Impact Statement

This paper presents work whose goal is to advance the field of Machine Learning. There are many potential societal consequences of our work, none which we feel must be specifically highlighted here.

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

Contributors, L. Lmdeploy: A toolkit for compressing, deploying, and serving llm, 2023.

Delimitrou, C. and Kozyrakis, C. Paragon: Qos-aware scheduling for heterogeneous datacenters. *Acm SIGPLAN Notices*, 48(4):77–88, 2013.

Delimitrou, C. and Kozyrakis, C. Quasar: Resource-efficient and qos-aware cluster management. *ACM Sigplan Notices*, 49(4):127–144, 2014.

Duan, J., Song, Z., Miao, X., Xi, X., Lin, D., Xu, H., Zhang, M., and Jia, Z. Parcae: Proactive,{Liveput-Optimized}{DNN} training on preemptible instances. In *21st USENIX Symposium on Networked Systems Design and Implementation (NSDI 24)*, pp. 1121–1139, 2024.

Dubey, A., Jauhri, A., Pandey, A., Kadian, A., Al-Dahle, A., Letman, A., Mathur, A., Schelten, A., Yang, A., Fan, A., et al. The llama 3 herd of models. *arXiv preprint arXiv:2407.21783*, 2024.

Gao, B., Song, F., Yang, Z., Cai, Z., Miao, Y., Dong, Q., Li, L., Ma, C., Chen, L., Xu, R., et al. Omni-math: A universal olympiad level mathematic benchmark for large language models. *arXiv preprint arXiv:2410.07985*, 2024.

GitHub. The world's most widely adopted ai developer tool, 2024. https://github.com/features/copilot.

Griggs, T., Liu, X., Yu, J., Kim, D., Chiang, W.-L., Cheung, A., and Stoica, I. M\'elange: Cost efficient large language model serving by exploiting gpu heterogeneity. *arXiv preprint arXiv:2404.14527*, 2024.

Gujarati, A., Karimi, R., Alzayat, S., Hao, W., Kaufmann, A., Vigfusson, Y., and Mace, J. Serving {DNNs} like clockwork: Performance predictability from the bottom up. In *14th USENIX Symposium on Operating Systems Design and Implementation (OSDI 20)*, pp. 443–462, 2020.

Hadi, M. U., Al Tashi, Q., Shah, A., Qureshi, R., Muneer, A., Irfan, M., Zafar, A., Shaikh, M. B., Akhtar, N., Wu, J., et al. Large language models: a comprehensive survey of its applications, challenges, limitations, and future prospects. *Authorea Preprints*, 2024.

Han, M., Zhang, H., Chen, R., and Chen, H. Microsecond-scale preemption for concurrent {GPU-accelerated}{DNN} inferences. In *16th USENIX Symposium on Operating Systems Design and Implementation (OSDI 22)*, pp. 539–558, 2022.

He, G., Jiang, Y., Xiao, W., Jiang, K., Wang, S., Wang, J., Du, Z., Jiang, Z., Zhang, X., Yuan, B., et al. Efficient pre-training of llms via topology-aware communication alignment on more than 9600 gpus. *arXiv preprint arXiv:2509.15940*, 2025.

Hendrycks, D., Burns, C., Basart, S., Zou, A., Mazeika, M., Song, D., and Steinhardt, J. Measuring massive multitask language understanding. *arXiv preprint arXiv:2009.03300*, 2020.

Huang, Y., Cheng, Y., Bapna, A., Firat, O., Chen, D., Chen, M., Lee, H., Ngiam, J., Le, Q. V., Wu, Y., et al. Gpipe: Efficient training of giant neural networks using pipeline parallelism. *Advances in neural information processing systems*, 32, 2019.

Isard, M., Prabhakaran, V., Currey, J., Wieder, U., Talwar, K., and Goldberg, A. Quincy: fair scheduling for distributed computing clusters. In *Proceedings of the ACM SIGOPS 22nd symposium on Operating systems principles*, pp. 261–276, 2009.

Jain, N., Han, K., Gu, A., Li, W.-D., Yan, F., Zhang, T., Wang, S., Solar-Lezama, A., Sen, K., and Stoica, I. Livecodebench: Holistic and contamination free evaluation of large language models for code. *arXiv preprint arXiv:2403.07974*, 2024.

Jaiswal, S., Jain, K., Simmhan, Y., Parayil, A., Mallick, A., Wang, R., Amant, R. S., Bansal, C., Ruhle, V., Kulkarni, A., et al. Sageserve: Optimizing llm serving on cloud data centers with forecast aware auto-scaling. *Proceedings of the ACM on Measurement and Analysis of Computing Systems*, 9(3):1–24, 2025a.

Jaiswal, S., Jain, K., Simmhan, Y., Parayil, A., Mallick, A., Wang, R., Amant, R. S., Bansal, C., Ruhle, V., Kulkarni, A., et al. Sageserve: Optimizing llm serving on cloud data centers with forecast aware auto-scaling. *Proceedings of the ACM on Measurement and Analysis of Computing Systems*, 9(3):1–24, 2025b.

Jeon, J. and Lee, S. Large language models in education: A focus on the complementary relationship between human teachers and chatgpt. *Education and Information Technologies*, 28(12):15873–15892, 2023.

Jiang, A. Q., Sablayrolles, A., Roux, A., Mensch, A., Savary, B., Bamford, C., Chaplot, D. S., Casas, D. d. l., Hanna, E. B., Bressand, F., et al. Mixtral of experts. *arXiv preprint arXiv:2401.04088*, 2024.

Jiang, Y., Gu, H., Lu, Y., and Yu, X. 2d-hra: Two-dimensional hierarchical ring-based all-reduce algorithm in large-scale distributed machine learning. *IEEE Access*, 8:183488–183494, 2020.

Jiang, Y., Fu, F., Miao, X., Nie, X., and Cui, B. Osdp: Optimal sharded data parallel for distributed deep learning. *arXiv preprint arXiv:2209.13258*, 2022.

Jiang, Y., Yan, R., Yao, X., Chen, B., and Yuan, B. Hexgen: Generative inference of foundation model over heterogeneous decentralized environment. *arXiv preprint arXiv:2311.11514*, 2023.

Jiang, Y., Fu, F., Yao, X., He, G., Miao, X., Klimovic, A., Cui, B., Yuan, B., and Yoneki, E. Demystifying cost-efficiency in llm serving over heterogeneous gpus. *arXiv preprint arXiv:2502.00722*, 2025a.

Jiang, Y., Fu, F., Yao, X., Wang, T., Cui, B., Klimovic, A., and Yoneki, E. Thunderserve: High-performance and cost-efficient llm serving in cloud environments. *arXiv preprint arXiv:2502.09334*, 2025b.

Jiang, Y., Fu, F., Zhao, W., Rabanser, S., Lane, N. D., and Yuan, B. Cascadia: A cascade serving system for large language models. *arXiv preprint arXiv:2506.04203*, 2025c.

Jiang, Y., Yan, R., and Yuan, B. Hexgen-2: Disaggregated generative inference of llms in heterogeneous environment. *arXiv preprint arXiv:2502.07903*, 2025d.

Jiang, Y., Fu, F., and Yoneki, E. Boute: Cost-efficient llm serving with heterogeneous llms and gpus via multi-objective bayesian optimization. *arXiv preprint arXiv:2602.10729*, 2026.

Kossmann, F., Fontaine, B., Khudia, D., Cafarella, M., and Madden, S. Is the gpu half-empty or half-full? practical scheduling techniques for llms. *arXiv preprint arXiv:2410.17840*, 2024.

Kwon, W., Li, Z., Zhuang, S., Sheng, Y., Zheng, L., Yu, C. H., Gonzalez, J., Zhang, H., and Stoica, I. Efficient memory management for large language model serving with pagedattention. In *Proceedings of the 29th Symposium on Operating Systems Principles*, pp. 611–626, 2023.

Li, B., Jiang, Y., Gadepally, V., and Tiwari, D. Llm inference serving: Survey of recent advances and opportunities. In *2024 IEEE High Performance Extreme Computing Conference (HPEC)*, pp. 1–8. IEEE, 2024.

Li, Z., Zheng, L., Zhong, Y., Liu, V., Sheng, Y., Jin, X., Huang, Y., Chen, Z., Zhang, H., Gonzalez, J. E., et al. {AlpaServe}: Statistical multiplexing with model parallelism for deep learning serving. In *17th USENIX Symposium on Operating Systems Design and Implementation (OSDI 23)*, pp. 663–679, 2023.

Lin, Y.-C., Kwon, W., Pineda, R., and Paravecino, F. N. Apex: An extensible and dynamism-aware simulator for automated parallel execution in llm serving. *arXiv preprint arXiv:2411.17651*, 2024.

Liu, Y., He, H., Han, T., Zhang, X., Liu, M., Tian, J., Zhang, Y., Wang, J., Gao, X., Zhong, T., et al. Understanding llms: A comprehensive overview from training to inference. *arXiv preprint arXiv:2401.02038*, 2024.

Liu, Z., Wang, J., Dao, T., Zhou, T., Yuan, B., Song, Z., Shrivastava, A., Zhang, C., Tian, Y., Re, C., et al. Deja vu: Contextual sparsity for efficient llms at inference time. In *International Conference on Machine Learning*, pp. 22137–22176. PMLR, 2023.

Ma, L., Van Aken, D., Hefny, A., Mezerhane, G., Pavlo, A., and Gordon, G. J. Query-based workload forecasting for self-driving database management systems. In *Proceedings of the 2018 International Conference on Management of Data*, pp. 631–645, 2018.

Mei, Y., Zhuang, Y., Miao, X., Yang, J., Jia, Z., and Vinayak, R. Helix: Distributed serving of large language models via max-flow on heterogeneous gpus. *arXiv preprint arXiv:2406.01566*, 2024.

Miao, X., Wang, Y., Jiang, Y., Shi, C., Nie, X., Zhang, H., and Cui, B. Galvatron: Efficient transformer training over multiple gpus using automatic parallelism. *arXiv preprint arXiv:2211.13878*, 2022.

Miao, X., Shi, C., Duan, J., Xi, X., Lin, D., Cui, B., and Jia, Z. Spotserve: Serving generative large language models on preemptible instances. In *Proceedings of the 29th ACM International Conference on Architectural Support for Programming Languages and Operating Systems, Volume 2*, pp. 1112–1127, 2024.

Miao, X., Oliaro, G., Zhang, Z., Cheng, X., Jin, H., Chen, T., and Jia, Z. Towards efficient generative large language model serving: A survey from algorithms to systems. *ACM Computing Surveys*, 58(1):1–37, 2025.

Naveed, H., Khan, A. U., Qiu, S., Saqib, M., Anwar, S., Usman, M., Akhtar, N., Barnes, N., and Mian, A. A comprehensive overview of large language models. *arXiv preprint arXiv:2307.06435*, 2023.

NVIDIA. ai-dynamo/dynamo: A datacenter-scale distributed inference serving framework, 2025. URL https://github.com/ai-dynamo/dynamo. GitHub repository.

Pan, Z., Wang, Y., Zhang, Y., Yang, S. B., Cheng, Y., Chen, P., Guo, C., Wen, Q., Tian, X., Dou, Y., et al. Magicscaler: Uncertainty-aware, predictive autoscaling. *Proceedings of the VLDB Endowment*, 16(12):3808–3821, 2023.

Patel, P., Chokse, E., Zhang, C., Shah, A., Goiri, Í., Maleki, S., and Bianchini, R. Splitwise: Efficient generative llm inference using phase splitting. In *2024 ACM/IEEE 51st Annual International Symposium on Computer Architecture (ISCA)*, pp. 118–132. IEEE, 2024.

Peng, C., Yang, X., Chen, A., Smith, K. E., PourNejatian, N., Costa, A. B., Martin, C., Flores, M. G., Zhang, Y., Magoc, T., et al. A study of generative large language model for medical research and healthcare. *NPJ digital medicine*, 6(1):210, 2023.

Peng, Y., Jiang, Y., Wang, C., and Yuan, B. Hexgen-text2sql: Optimizing llm inference request scheduling for agentic text-to-sql workflow. *arXiv preprint arXiv:2505.05286*, 2025.

Qiao, Y., Anzai, S., Yu, S., Ma, H., Yang, S., Wang, Y., Kim, M., Wu, Y., Zhou, Y., Xing, J., et al. Conserve: Fine-grained gpu harvesting for llm online and offline co-serving. *arXiv preprint arXiv:2410.01228*, 2024.

Reid, M., Savinov, N., Teplyashin, D., Lepikhin, D., Lillicrap, T., Alayrac, J.-b., Soricut, R., Lazaridou, A., Firat, O., Schrittwieser, J., et al. Gemini 1.5: Unlocking multimodal understanding across millions of tokens of context. *arXiv preprint arXiv:2403.05530*, 2024.

Schwarzkopf, M., Konwinski, A., Abd-El-Malek, M., and Wilkes, J. Omega: flexible, scalable schedulers for large compute clusters. In *Proceedings of the 8th ACM European Conference on Computer Systems*, pp. 351–364, 2013.

Shoeybi, M., Patwary, M., Puri, R., LeGresley, P., Casper, J., and Catanzaro, B. Megatron-lm: Training multibillion parameter language models using model parallelism. *arXiv preprint arXiv:1909.08053*, 2019.

Stojkovic, J., Zhang, C., Goiri, Í., Torrellas, J., and Choukse, E. Dynamollm: Designing llm inference clusters for performance and energy efficiency. *arXiv preprint arXiv:2408.00741*, 2024.

Sun, B., Huang, Z., Zhao, H., Xiao, W., Zhang, X., Li, Y., and Lin, W. Llumnix: Dynamic scheduling for large language model serving. *arXiv preprint arXiv:2406.03243*, 2024.

Tong, C., Jiang, Y., Chen, G., Zhao, T., Lu, S., Qu, W., Yang, E., Ai, L., and Yuan, B. Parallax: Efficient llm inference service over decentralized environment. *arXiv preprint arXiv:2509.26182*, 2025.

Touvron, H., Martin, L., Stone, K., Albert, P., Almahairi, A., Babaei, Y., Bashlykov, N., Batra, S., Bhargava, P., Bhosale, S., et al. Llama 2: Open foundation and fine-tuned chat models. *arXiv preprint arXiv:2307.09288*, 2023.

Vaswani, A. Attention is all you need. *Advances in Neural Information Processing Systems*, 2017.

Wang, Y., Chen, Y., Li, Z., Kang, X., Tang, Z., He, X., Guo, R., Wang, X., Wang, Q., Zhou, A. C., et al. Burstgpt: A real-world workload dataset to optimize llm serving systems. 2024a.

Wang, Y., Jiang, Y., Miao, X., Fu, F., Zhu, S., Nie, X., Tu, Y., and Cui, B. Improving automatic parallel training via balanced memory workload optimization. *IEEE Transactions on Knowledge and Data Engineering*, 36 (8):3906–3920, 2024b.

Wang, Y., Chen, Y., Li, Z., Kang, X., Fang, Y., Zhou, Y., Zheng, Y., Tang, Z., He, X., Guo, R., et al. Burstgpt: A real-world workload dataset to optimize llm serving systems. In *Proceedings of the 31st ACM SIGKDD Conference on Knowledge Discovery and Data Mining V. 2*, pp. 5831–5841, 2025a.

Wang, Y., Jiang, Y., Cui, B., and Fu, F. Thinking short and right over thinking long: Serving llm reasoning efficiently and accurately. *arXiv preprint arXiv:2505.13326*, 2025b.

Wu, B., Zhong, Y., Zhang, Z., Huang, G., Liu, X., and Jin, X. Fast distributed inference serving for large language models. *arXiv preprint arXiv:2305.05920*, 2023.

Xiong, D., Chen, L., Jiang, Y., Li, D., Wang, S., and Wang, S. Revisiting the time cost model of allreduce. *arXiv preprint arXiv:2409.04202*, 2024.

Yan, R., Jiang, Y., Chen, Z., Mai, H., Chen, B., and Yuan, B. Fsa: An alternative efficient implementation of native sparse attention kernel. *arXiv preprint arXiv:2508.18224*, 2025a.

Yan, R., Jiang, Y., Wu, T., Gao, J., Mei, Z., Fu, W., Mai, H., Wang, W., Wu, Y., and Yuan, B. Areal-hex: Accommodating asynchronous rl training over heterogeneous gpus. *arXiv preprint arXiv:2511.00796*, 2025b.

Yu, G.-I., Jeong, J. S., Kim, G.-W., Kim, S., and Chun, B.-G. Orca: A distributed serving system for {Transformer-Based} generative models. In *16th USENIX Symposium on Operating Systems Design and Implementation (OSDI 22)*, pp. 521–538, 2022.

Yu, Y., Si, X., Hu, C., and Zhang, J. A review of recurrent neural networks: Lstm cells and network architectures. *Neural computation*, 31(7):1235–1270, 2019.

Zhang, H., Tang, Y., Khandelwal, A., and Stoica, I. {SHEPHERD}: Serving {DNNs} in the wild. In *20th USENIX Symposium on Networked Systems Design and Implementation (NSDI 23)*, pp. 787–808, 2023.

Zhang, J., Su, R., Liu, C., Wei, J., Wang, Z., Zhang, P., Wang, H., Jiang, H., Huang, H., Xiang, C., et al. A survey of efficient attention methods: Hardware-efficient, sparse, compact, and linear attention.

Zhang, J., Zeng, Y., and Starly, B. Recurrent neural networks with long term temporal dependencies in machine tool wear diagnosis and prognosis. *SN Applied Sciences*, 3(4):442, 2021.

Zhang, L., Jiang, Y., He, G., Chen, X., Lv, H., Yao, Q., Fu, F., and Chen, K. Efficient mixed-precision large language model inference with turbomind. *arXiv preprint arXiv:2508.15601*, 2025.

Zhang, S., Roller, S., Goyal, N., Artetxe, M., Chen, M., Chen, S., Dewan, C., Diab, M., Li, X., Lin, X. V., et al. Opt: Open pre-trained transformer language models. *arXiv preprint arXiv:2205.01068*, 2022.

Zhao, W., Ren, X., Hessel, J., Cardie, C., Choi, Y., and Deng, Y. Wildchat: 1m chatgpt interaction logs in the wild. *arXiv preprint arXiv:2405.01470*, 2024a.

Zhao, Y., Lin, C.-Y., Zhu, K., Ye, Z., Chen, L., Zheng, S., Ceze, L., Krishnamurthy, A., Chen, T., and Kasikci, B. Atom: Low-bit quantization for efficient and accurate llm serving. *Proceedings of Machine Learning and Systems*, 6:196–209, 2024b.

Zhao, Y., Yang, S., Zhu, K., Zheng, L., Kasikci, B., Zhou, Y., Xing, J., and Stoica, I. Blendserve: Optimizing offline inference for auto-regressive large models with resource-aware batching. *arXiv preprint arXiv:2411.16102*, 2024c.

Zheng, L., Li, Z., Zhang, H., Zhuang, Y., Chen, Z., Huang, Y., Wang, Y., Xu, Y., Zhuo, D., Xing, E. P., et al. Alpa: Automating inter-and {Intra-Operator} parallelism for distributed deep learning. In *16th USENIX Symposium on Operating Systems Design and Implementation (OSDI 22)*, pp. 559–578, 2022.

Zheng, L., Chiang, W.-L., Sheng, Y., Li, T., Zhuang, S., Wu, Z., Zhuang, Y., Li, Z., Lin, Z., Xing, E. P., et al. Lmsys-chat-1m: A large-scale real-world llm conversation dataset. *arXiv preprint arXiv:2309.11998*, 2023.

Zhong, Y., Liu, S., Chen, J., Hu, J., Zhu, Y., Liu, X., Jin, X., and Zhang, H. Distserve: Disaggregating prefill and decoding for goodput-optimized large language model serving. *arXiv preprint arXiv:2401.09670*, 2024.

Zhou, Z., Wei, X., Zhang, J., and Sun, G. {PetS}: A unified framework for {Parameter-Efficient} transformers serving. In *2022 USENIX Annual Technical Conference (USENIX ATC 22)*, pp. 489–504, 2022.

# A. Extended Related Work

**LLM inference serving.** There are plenty of recent researches focused on optimizing LLM inference and serving (Li et al., 2023; Kwon et al., 2023; Agrawal et al., 2024b; Liu et al., 2023; Wu et al., 2023; Zhou et al., 2022; Yu et al., 2022; Jiang et al., 2023; Contributors, 2023; Zhang et al., 2025; Jiang et al., 2025c; Wang et al., 2025b; Yan et al., 2025a; Zhang et al.). AlpaServe (Li et al., 2023) adopts model parallelism to optimize LLM serving performance. SARATHI (Agrawal et al., 2024b) introduces a chunked-prefill approach and piggybacks decoding requests to improve hardware utilization. Splitwise (Patel et al., 2024) splits the prefill and decoding phases onto separate machines to optimize hardware utilization. SpotServe (Miao et al., 2024) supports LLM inference using preemptible instances for improving cost efficiency. HexGen (Jiang et al., 2023) proposes asymmetric parallelism and an advanced scheduling algorithm to deploy generative inference in heterogeneous environments. We plan to explore workload-aware LLM inference with disaggregated inference architecture and heterogeneous GPU environments in future work.

**Request scheduling in model inference.** Numerous systems have been developed to optimize request scheduling for model inference serving (dee, 2023; Gujarati et al., 2020; Li et al., 2023; Han et al., 2022; Zhang et al., 2023; Sun et al., 2024; Patel et al., 2024; Kossmann et al., 2024; Peng et al., 2025). Among them, DeepSpeed-MII (dee, 2023) uses a round-robin dispatching policy to arrange multi-instance LLM serving. Llumnix (Sun et al., 2024) integrates dynamic request migration to ensure high throughput and low latency LLM serving, provides SLO for prioritized requests, and auto-scales instances for resource efficiency with a unified load-aware dynamic scheduling policy. (Kossmann et al., 2024) prioritizes request according to their anticipated memory demand and the current system load, quantifies the load of each server, and routes request to server with the lowest load. Differently, OSERVE distinguishes requests by type, co-optimizes request scheduling with resource allocations and parallel strategies of model replicas, and directs requests to the most suitable replicas to maximize resource utilization.

**Job scheduling in clusters.** There is also a line of research that considers the job scheduling in clusters (Isard et al., 2009; Schwarzkopf et al., 2013; Delimitrou & Kozyrakis, 2013; 2014; Jiang et al., 2020; Xiong et al., 2024). However, our work focuses on the request scheduling for LLM serving, which has a different goal.

**Hybrid model parallelism.** Hybrid model parallelism combines tensor parallelism with pipeline parallelism to efficiently scale LLM training and inference beyond what either strategy achieves alone (Zheng et al., 2022; Li et al., 2023; Miao et al., 2022; Jiang et al., 2022; Wang et al., 2024b; He et al., 2025; Yan et al., 2025b). Hybrid parallelism enables fitting massive models across multiple GPUs while minimizing inter-machine communication overhead and latency, which is critical for meeting real-time inference requirements.

# B. Extended Discussion

**Scheduling optimality.** Some of the proposed algorithms are developed based on heuristics (§4 and §5.2), which may introduce sub-optimality to the scheduling results (i.e., serving strategies and switch plans). However, this increases the efficiency of the algorithms by a large extent, and meanwhile the scheduling results are still effective, as evaluated in §6.4. Consequently, although our heuristics trade off theoretical optimality for the search efficiency, we believe they are well-suited for practical deployment in dynamic serving environments, since they ensure the algorithms generate near-optimal plans under real-time constraints.

**Fault tolerance and serving with spot instances.** Employing heterogeneous model deployment increases the complexity to manage the model replicas, which may raise hurdles for fault tolerance. Nevertheless, our scheduling algorithm's flexibility makes it well-suited for extension to fault-tolerant and spot instance scenarios — upon detecting a failure or resource change, the algorithm can initiate a rerun with an updated cluster size, identify the optimal model deployment, and seamlessly adjust through model deployment switching.

**Extensibility to heterogeneous GPU types.** Many recent studies have focused on cost-efficient LLM serving using heterogeneous GPU resources (Jiang et al., 2023; Mei et al., 2024; Miao et al., 2024; Griggs et al., 2024; Patel et al., 2024; Jiang et al., 2025a;d;b; Tong et al., 2025; Jiang et al., 2026). However, adapting OSERVE to heterogeneous environments presents significant challenges, as the search space expands dramatically when accounting for the varying computational capabilities, memory bandwidths, and memory limits of different GPU types. We leave this extension for future exploration.

## C. Simple Example

**A simple example.** To further motivate our problem formulation, consider a cluster with 8 GPUs serving two distinct workload types. Suppose there are 100 incoming requests of workload type 1 ($\lambda_1 = 100$) and 50 incoming requests of workload type 2 ($\lambda_2 = 50$). Let $C_{k,j}$ denote the processing rate (in requests per second) of the $k$-th model replica on the $j$-th workload, and let $f_{k,j}$ represent the fraction of workload $j$ assigned to replica $k$. As shown in Figure 4, we consider three cases:

*Case 1:* Assume a model deployment consisting of two identical replicas, each configured with ($T_{1,2} = 2, P_{1,2} = 2$), and each offering rates $C_{\sim,1} = 10$ requests/s and $C_{\sim,2} = 5$ requests/s. If workload type 1 is directed entirely to the first replica ($f_{1,1} = 1$) and workload type 2 is directed entirely to the second replica ($f_{2,2} = 1$), the total completion time is at least $\max(\lambda_1 f_{1,1}/C_{1,1}, \lambda_2 f_{2,2}/C_{2,2}) = 20$ seconds.

*Case 2:* Now consider a different model configuration with three replicas, where the first replica has ($T_1 = 2, P_1 = 2$) and the second and third replicas have ($T_{2,3} = 2, P_{2,3} = 1$). Under these settings, replicas 2 and 3 each process the workloads at a rate of $C_{\sim,1} = 5$ for workload 1 and $C_{\sim,2} = 3$ for workload 2. Assigning all of workload type 1 to the first replica ($f_{1,1} = 1$) and splitting workload type 2 evenly across replicas 2 and 3 ($f_{2,2} = 0.5$, $f_{3,2} = 0.5$) reduces the completion time to 16.67 seconds. This improvement comes from more appropriate model deployment.

*Case 3:* Using the same model deployment as in Case 2, we can further optimize workload assignment. Suppose we route all of workload type 1 and 18% of workload type 2 to the first replica ($f_{1,1} = 1$, $f_{1,2} = 0.18$), and distribute the remaining 82% of workload type 2 evenly between replicas 2 and 3 ($f_{2,2} = 0.41$, $f_{3,2} = 0.41$). By carefully balancing the fractions of requests, the completion time decreases to approximately 13.67 seconds. This improvement comes from more appropriate workload assignment.

These cases highlight the importance of jointly optimizing model deployments and workload assignments. By comparing the outcomes—20 seconds with a simple two-replica scheme, 16.67 seconds with a three-replica scheme and straightforward routing, and 13.67 seconds through careful fraction allocations—we demonstrate that more nuanced model deployment and workload assignment can substantially improve system throughput and efficiency.

## D. One-Time Profiling

We use a one-time profiling strategy (similar to previous works (Patel et al., 2024; Jaiswal et al., 2025a; Lin et al., 2024)) to evaluate model capacity under different parallelism strategies for various workload types (This approach is based on the profiling method used in Vidur (Agrawal et al., 2024a)), which captures the following components:

- **Inference-prefilling latency**: the latency of a single transformer layer across varying tensor-parallel (TP) degrees and workload types.

- **Inference-decoding latency**: the decoding latency of a single transformer layer under the same TP- and workload-type variations.

- **Pipeline communication latency**: the communication latency between GPUs for various workload types.

Using these measurements, we estimate per-request latency for any configuration by combining each layer's TP costs (both computation and communication) with the pipeline-parallelism (PP) communication cost. When estimating throughput, we treat the prefill and decoding phases separately:

- **Prefill phase**: compute-bound, with batched processing capacity determined by the sum of individual layer latencies.

- **Decoding phase**: memory-bound, with batched processing capacity defined by a single latency value.

This distinction has been validated in several studies (Zhong et al., 2024; Patel et al., 2024).

## E. Flow Network Guided Model Deployment Generation

The Flow Network Guided Generation Algorithm (§4.3) focuses on optimizing cluster-wide model deployments by iteratively adjusting GPU allocations and model parallelization strategies. Starting with an initial distribution of GPUs across multiple

replicas and a chosen strategy for each replica, the algorithm uses the lower-level flow network (§4.2) to measure how well resources are utilized. From the generated flow assignments, it identifies overutilized replicas—those operating at full capacity—and underutilized replicas—those operating at low capacity. In response, the algorithm adaptively merges, splits, or swaps GPU resources among replicas, seeking to balance the load and improve the overall throughput.

In particular, overutilized replicas may merge with one another to form more efficient configurations or receive additional GPUs from underutilized replicas. Conversely, underutilized replicas might be split to form new groups or relinquish some of their GPUs to support overutilized replicas. By repeatedly adjusting these allocations and evaluating multiple parallel strategies, the algorithm converges toward an effective configuration that improves throughput. This method ensures a more dynamic and informed approach to resource management in GPU clusters, achieving higher performance and more efficient utilization than a static or one-shot allocation strategy. We demonstrate the algorithm pseudo-code in Algorithm 1. The detailed steps are as follow:

**Initialization.** The algorithm starts with a random uniform initial model deployment (with uniform resource allocation and parallel strategies), specifying how GPUs are allocated to each model replica and which parallelization strategy each replica uses. A record of the original configuration is stored at the beginning of each iteration to allow rollback if no improvement is found.

**Lower-level flow network evaluation.** In each iteration, the algorithm invokes the FlowNetwork function $\mathcal{L}$, which takes the current allocation and strategies $\{d_r\}$ and $\{s_r\}$ as inputs and returns a flow assignment and an achievable throughput. Based on the flow assignment, replicas are categorized as either overutilized (those operating at full capacity) or underutilized (those operating below capacity).

**Adaptive resource adjustment.** Overutilized replicas can either `merge` with another overutilized replica, combining their GPU allocations, or `swap` GPUs with an underutilized replica to balance load. Underutilized replicas can either be `split` into two new replicas (to potentially better match resource requirements) or `swap` GPUs with an overutilized replica. Note that these mutation operations (`merge`, `split`, `swap`) are chosen randomly in each iteration, which helps keep the search unbiased and versatile.

**Strategy evaluation and reversion.** After adjusting GPU allocations, the algorithm explores all possible parallelization strategy combinations $\{s_r'\}$ to find the one that yields the highest throughput. If the new best strategy combination leads to a higher throughput than the current recorded best, the changes are accepted. Otherwise, the algorithm reverts to the previously recorded configuration, ensuring that detrimental adjustments are discarded.

**Convergence and iteration.** This process repeats until the algorithm converges (e.g., no further improvements are found) or a maximum number of iterations is reached. Upon termination, the final $\{d_r\}$ and $\{s_r\}$ represent a model deployment that maximizes throughput according to the given constraints and cluster conditions.

In essence, the algorithm dynamically negotiates the interplay between GPU distribution and parallelization strategies, driven by insights from the flow network, to elevate the cluster's throughput and efficiency. Additional experimental results about this algorithm are demonstrated in §6.4. The code snippets for this algorithm are available at this URL.

## F. Greedy Algorithm for Ad Hoc Model Switching

The Greedy Algorithm (§5.2) is designed to efficiently determine how to transfer large-scale model parameters between GPUs in order to reconfigure model deployments. Given a set of parameters, their source devices (where they are currently stored), and target devices (where they are needed), the algorithm constructs a switch plan that specifies which source device should send each required parameter shard to which target device. By initializing zeroed communication loads and incrementally adding parameter shards to minimize the communication overhead, the algorithm ensures that each parameter shard assignment is made by choosing the source device with the lowest current data transfer volume. This method inherently balances the communication load, striving to avoid bottlenecks and latency spikes.

A key enhancement to the algorithm's efficiency lies in prioritizing intra-machine communication—those transfers that occur within the same machine and thus can leverage ultra-high-speed GPU interconnects (e.g., NVLink)—over slower inter-machine communication. By first attempting to satisfy parameter requirements using intra-machine sources, the

*Table 3.* Resource allocations and strategies at P1-6 for trace 1 and trace 2 while serving OPT-66B models on 16 GPUs.

| Trace | Span | DP | Allocations and Strategies |
|---|---|---|---|
| Trace 1 | P1 | 5 | (TP=3, PP=2), (TP=2, PP=2), (TP=2)×3 |
| | P2 | 4 | (TP=4, PP=2), (TP=2, PP=2), (TP=2)×2 |
| | P3 | 4 | (PP=4), (TP=2, PP=2)×3 |
| | P4 | 4 | (TP=3, PP=2), (TP=2), (TP=2, PP=2)×2 |
| | P5 | 3 | (TP=4, PP=2), (TP=2, PP=2)×2 |
| | P6 | 3 | (TP=2, PP=2), (TP=3, PP=2)×2 |
| Trace 2 | P1 | 4 | (TP=3, PP=2), (TP=2), (TP=2, PP=2)×2 |
| | P2 | 3 | (TP=4, PP=2), (TP=3, PP=2), (TP=2) |
| | P3 | 4 | (TP=3, PP=2), (TP=2), (TP=2, PP=2)×2 |
| | P4 | 3 | (TP=4, PP=2), (TP=2, PP=2)×2 |
| | P5 | 4 | (TP=4, PP=2), (TP=2, PP=2), (TP=2)×2 |
| | P6 | 5 | (TP=4, PP=2), (PP=2), (TP=2)×3 |

algorithm reduces overall data transfer time and network congestion. While not guaranteed to be absolutely optimal, this greedy approach provides a highly practical solution for large-scale GPU clusters, delivering near-optimal load balancing and improved parameter switching performance. We demonstrate the algorithm pseudo-code in Algorithm 2. The detailed steps are as follow:

**Initialization.** The algorithm starts with an empty switch plan $P$ It maintains a communication load counter $C_{s \to t}$, for every source-target pair $(s, t)$, initially zero.

**Model shard assignment.** For each model shard $m$, the algorithm identifies its source devices $S_m$ (devices that currently hold $m$) and its target devices $T_m$ (devices that need $m$). For each target device $t \in T_m$, if the target already has the model shard $m$, no transfer is needed. Otherwise, the algorithm proceeds to select a suitable source device.

**Intra- and inter-machine preference.** The source devices $S_m$ are partitioned into two subsets: $S_m^{(\text{intra})}$ for devices within the same machine as $t$ and $S_m^{(\text{inter})}$ for devices located on other machines. The algorithm first attempts to choose a source from $S_m^{(\text{intra})}$ to exploit higher intra-machine bandwidth. If no intra-machine source is available, it falls back to selecting from $S_m^{(\text{inter})}$.

**Greedy selection.** The source device $s^*$ is chosen by finding the one with the minimum existing communication load $C_{s \to t}$. By always choosing the currently lightest-loaded source device, the algorithm spreads out data transfers and avoids creating communication hotspots. Once $s^*$ is chosen, $C_{s^* \to t}$ is updated by adding the volume $V_{m,t}$ of the transferred parameter shard, and the mapping $(m, s^*, t)$ is added to the switch plan $P$.

**Outcome.** After processing all required model shards, $P$ defines a communication plan that aims to minimize the overall data transfer cost. While not provably optimal, the algorithm is a practical solution for complex, large-scale GPU clusters, providing a near-optimal communication schedule with relatively little overhead.

This method can be seamlessly integrated into larger systems where rapid reconfiguration of model replicas is crucial, ensuring quick adaptation to workload changes and resource constraints, and leading to improved overall system throughput and responsiveness. Additional experimental results about the effect of ad hoc model switching and this algorithm are demonstrated in §6.3 and §6.4. The code snippets for this algorithm are available at this URL.

# G. Case Study

**Resource allocations and parallel strategies over time.** To illustrate how allocations and strategies of OSERVE change over time, we benchmarked OSERVE's resource allocations and parallel strategies across time spans P1–P6 from trace 1 and trace 2 while serving OPT-66B models on 16 GPUs. As shown in Table 3, the resource allocations and parallel strategies gradually change as workloads evolve.

---

**Algorithm 1** Flow Network Guided Model Deployment Generation

---

**Input:**

Total GPUs $D$, number of replicas $R$, parallel strategy set $\mathcal{S}$, FlowNetwork function $\mathcal{L}$, capacities $cap(\cdot)$.

**Output:**

A model deployment $(\{d_r\}, \{s_r\})$.

**Initialization:**

 A random uniform model deployment $(\{d_r\}, \{s_r\})$.

**repeat**

     `// Record current configuration`

     $\{d_r^{orig}\} \leftarrow \{d_r\}, \{s_r^{orig}\} \leftarrow \{s_r\}$

         `// Flow network optimization`

     $(flow\_assignment, throughput) \leftarrow \mathcal{L}(\{d_r\}, \{s_r\})$;

         `// Overutilized model replicas`

     $O \leftarrow \{r \mid flow\_assignment(r) = cap(r)\}$

         `// Underutilized model replicas`

     $U \leftarrow \{r \mid flow\_assignment(r) < cap(r)\}$

     **foreach** $r \in O$ **do**

         $op \leftarrow$ SelectOperation(`merge`, `swap`)

         **if** $op = $ `merge` **and** $|O| > 1$ **then**

             `// Merge two replicas into one`

             Choose $r' \in O, r' \neq r$;

             $d_r \leftarrow d_r + d_{r'}$; remove $r'$.

         **else if** $op = $ `swap` **and** $U \neq \emptyset$ **then**

             `// Swap GPUs between replicas`

             Choose $u \in U$ and $\delta$;

             $d_r \leftarrow d_r + \delta, d_u \leftarrow d_u - \delta$.

     **foreach** $r \in U$ **do**

         $op \leftarrow$ SelectOperation(`split`, `swap`)

         **if** $op = $ `split` **then**

             `// Split one replica into two`

             Split $d_r$ into $(d_{r_1}, d_{r_2})$;

             replace $r$ with $r_1, r_2$.

         **else if** $op = $ `swap` **and** $O \neq \emptyset$ **then**

             `// Swap GPUs between replicas`

             Choose $o \in O$ and $\delta$;

             $d_o \leftarrow d_o + \delta, d_r \leftarrow d_r - \delta$.

     `// Find the strategy combination that maximizes throughput`

     $(s_r', throughput') \leftarrow \arg\max_{\{s_r'\} \in \mathcal{S}^R} throughput(\mathcal{L}(\{d_r\}, \{s_r'\}))$

     **if** $throughput' > throughput$ **then**

         $\{s_r\} \leftarrow \{s_r'\}$

         $throughput \leftarrow throughput'$

     **else**

         $\{d_r\} \leftarrow \{d_r^{orig}\}, \{s_r\} \leftarrow \{s_r^{orig}\}$

**until** *convergence or max iterations*;

**return** $(\{d_r\}, \{s_r\})$

---

---

**Algorithm 2** Greedy Switch Plan Algorithm

---

**Input:**

Set of model parameters $M$; For each model shard $m \in M$: a set of source devices $S_m$ that currently hold $m$, and a set of target devices $T_m$ that require $m$; Parameter volume $V_{m,t}$ for each required pair $(m, t)$; Intra-machine bandwidth priority.

**Output:**

A switch plan $P$ mapping each required model shard $(m, t)$ to a source device $s$.

**Initialization:**

Set $P \leftarrow \emptyset$. Set $C_{s \rightarrow t} \leftarrow 0$ for all $(s, t)$ pairs.

**foreach** $m \in M$ **do**

 Determine $S_m$ (source devices holding $m$).

 Determine $T_m$ (target devices requiring $m$).

 **foreach** $t \in T_m$ **do**

  **if** $t \notin S_m$ **then**

   // Only proceed if $t$ does not already hold $m$

   Partition $S_m$ into $S_m^{(\text{intra})}$ and $S_m^{(\text{inter})}$ based on intra- or inter-machine placement relative to $t$.

   **if** $S_m^{(\text{intra})} \neq \emptyset$ **then**

    $s^* \leftarrow \arg\min_{s \in S_m^{(\text{intra})}} C_{s \rightarrow t}$

   **else**

    $s^* \leftarrow \arg\min_{s \in S_m^{(\text{inter})}} C_{s \rightarrow t}$

   Update $C_{s^* \rightarrow t} \leftarrow C_{s^* \rightarrow t} + V_{m,t}$.

   Update $P \leftarrow P \cup \{(m, s^*, t)\}$.

**return** $P$

---

Specifically, in trace 1, during the transition from P1 to P2, two model replicas configured as (TP=3, PP=2) and (TP=2) are merged into a single replica with the configuration (TP=4, PP=2). This adjustment is driven by an increase in the proportion of workload types 1 and 2, coupled with a decrease in types 3 and 4. Compared to previous strategies, the (TP=4, PP=2) configuration improves the system's average request processing latency by approximately 20%, thereby enhancing model serving efficiency. As the workload trend persists, the scheduling algorithm gradually allocates more GPUs to individual model replicas (i.e., increases the MP size) and reduces the number of model replicas (i.e., decreases the DP size) to adapt to the changing workload. Similarly, although trace 2 presents a different workload fluctuation trend from P1 to P6, OSERVE adjusts its resource allocations and parallel strategies accordingly based on the workload changes, thereby enhancing system efficiency.

## H. Compare with Llumnix and Dynamo+vLLM

**Comparison with Llumnix.** We also compare OSERVE with Llumnix, a state-of-the-art inference framework that integrates dynamic request migration to handle workload fluctuations. Figure 17 shows the comparison—when serving the Llama-30B and Llama2-70B models with 8 and 16 GPUs across different traces, OSERVE outperforms Llumnix by 1.32-1.51× in P99 latency and throughput. Llumnix does not account for the impact of different model deployments (i.e., resource allocations and parallelism strategies) on LLM serving across various workload types, limiting its ability to fully utilize system resources. In contrast, OSERVE co-optimizes model deployment with request scheduling, leading to significant performance improvements.

**Comparison with Dynamo+vLLM.** As shown in Figure 18, we compare OSERVE with Dynamo+vLLM, which autoscaling prefill/decoding workers and routes requests via KV-aware scheduling. OSERVE improves end-to-end system performance by 12–20% when serving Llama-30B and Llama2-70B on 8-16 GPUs across multiple traces. The gains stem from dynamically reconfiguring parallelism to match workload mix, whereas Dynamo fixes per-worker parallelism, overlooking the parallelism–workload interaction. These results highlight the benefit of OSERVE 's co-optimization of request scheduling and runtime parallelism.

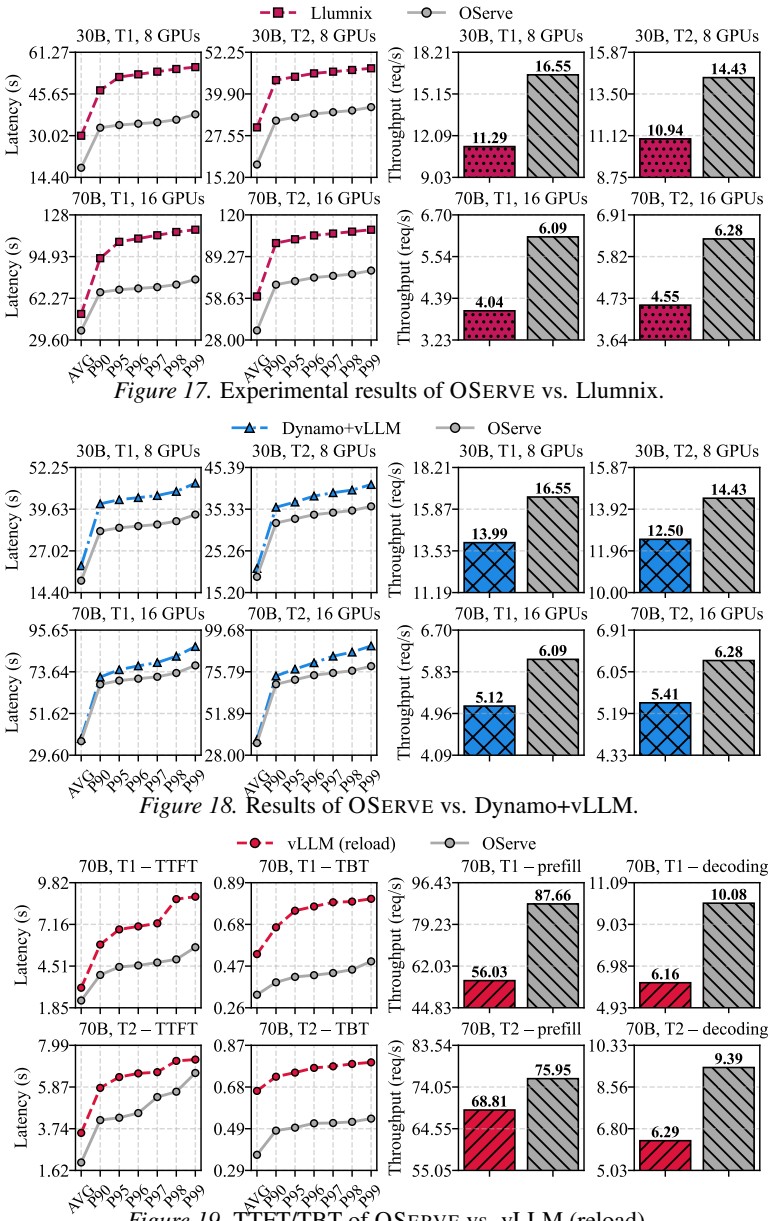

*Figure 17.* Experimental results of OSERVE vs. Llumnix.

*Figure 18.* Results of OSERVE vs. Dynamo+vLLM.

*Figure 19.* TTFT/TBT of OSERVE vs. vLLM (reload).

## I. TTFT and TBT Results

**TTFT and TBT results.** To further demonstrate the effectiveness of OSERVE in online serving scenarios, we benchmarked its TTFT (Time-To-First-Token, i.e., time to generate the first token) and TBT (Time-Between-Token, i.e., average generation time per token) against those of vLLM across various traces. As shown in Figure 19, OSERVE achieves 1.1–1.6× improvements in P99 TTFT latency and 1.5–1.6× improvements in P99 TBT latency, demonstrating that, in addition to reducing end-to-end latency, OSERVE also significantly enhances other key serving metrics.

## J. Experiments on Homogeneous Workloads

**Runtime overhead on homogeneous workloads.** To verify that OSERVE does not introduce unnecessary overhead when workload heterogeneity is absent, we evaluate on a synthetic homogeneous trace with uniform request lengths and stable arrival rates. Under these conditions, OSERVE's scheduler converges to a near-uniform deployment within 2-3 iterations, achieving P99 latency within 3% of vLLM (static) and comparable throughput. This demonstrates that OSERVE gracefully degrades to existing baselines when workload separability is low.

## K. Additional Discussion

**Extensibility to emerging parallelisms.** The parallelism strategy optimization (§4.3) is designed to be extensible. Additional parallelisms such as Expert Parallelism (EP) for MoE models and Sequence Parallelism (SP) for long-context workloads can be incorporated into the enumeration and optimization process without fundamental changes to the scheduling algorithm. This requires only profiling the new strategy's capacity characteristics and adding it to the search space.

