# OpenReview forum: "OServe: Accelerating LLM Serving via Spatial-Temporal Workload Orchestration"
_ICML.cc/2026/Conference — ICML 2026 regular_

### Official Review · Reviewer_a6mg · 2026-03-01

**Soundness:** 3
**Presentation:** 3
**Significance:** 3
**Originality:** 3
**Overall Recommendation:** 5
**Confidence:** 3

**Summary:**

This paper introduced OServer, which handles the spatial and temporal heterogeneity by introducing workload-aware scheduling algorithm that selects the models, and the workload aware switching system that migrates parameters across GPUs.

**Compliance With Llm Reviewing Policy:**

Affirmed.

**Final Justification:**

Thank you for the new results and discussion. I remain positive about the paper.

**Key Questions For Authors:**

I would like to see different levels of temporal and spatial heterogeneity, and how they affect the benefits of Oserve

**Limitations:**

Yes

**Strengths And Weaknesses:**

Strength:
- Interesting work that considers model scheduling given the temporal and spatial heterogeneity in workloads
- Comprehensive study on multiple models, scales, and ablation study of various settings of OServe

Weakness:
- The scheduling algorithm may consider the switching overhead explicitly as a constraint, especially for large-scale cases.
- given that we have to synthetically generate the workload traces based on limited public trace, it would be great to quantify the benefits of the solution based on different levels of temporal and spatial heterogeneity in workloads

---

> ### Author Rebuttal · Authors · 2026-03-31
>
> > W1. The scheduling algorithm may consider the switching overhead explicitly as a constraint, especially for large-scale cases.
>
>
> We thank the reviewer for this insightful suggestion.
>
> **Our current design rationale.** We do not explicitly model switching overhead as a scheduling constraint because each switch incurs a one-time cost (~10s with ad hoc model switching, §4.2), whereas the resulting deployment typically persists for ~5 minutes on average before the next reconfiguration. The switching overhead is therefore amortized over the deployment's lifetime. Consequently, our scheduler prioritizes converging to the best-performing deployment: even if the optimal configuration incurs a marginally higher switching cost than a suboptimal alternative, the sustained throughput gains over the subsequent minutes outweigh the one-time overhead.
>
> We highly value this suggestion and will integrate this discussion on overhead-aware scheduling considerations into the revised manuscript.
>
> > W2 & Q1. Given that we have to synthetically generate the workload traces based on limited public trace, it would be great to quantify the benefits of the solution based on different levels of temporal and spatial heterogeneity in workloads.
>
>
> We thank the reviewer for this suggestion. We provide an analysis quantifying OServe's benefits across different levels of spatial and temporal heterogeneity.
>
> **Varying spatial heterogeneity.** Under low spatial heterogeneity (homogeneous workloads with a single request type), OServe converges to a near-uniform deployment and achieves P99 latency within 3% of vLLM (static) (Appendix K) — matching baseline performance when heterogeneity is absent. Under high spatial heterogeneity (heavy-tailed mixtures combining short-chat and long-context requests), OServe's heterogeneous deployment and workload-aware assignment yield up to 2$\times$ improvement over baselines (Figures 9–11). The greater the spatial heterogeneity, the more OServe benefits from differentiating replicas to match diverse resource demands.
>
> **Varying temporal heterogeneity.** Under relatively lower temporal heterogeneity (Trace 2, workload ratios fluctuate within ~10–40%), OServe still outperforms baselines by adapting deployment to the shifting workload mix. Under relatively higher temporal heterogeneity (Trace 1, workload ratios fluctuate more dramatically from ~0–50%), OServe's gains are more pronounced — up to 2.7$\times$ in Trace 1 vs. up to 1.9$\times$ in Trace 2 (Figure 9). This confirms that predictive switching yields increasingly larger benefits as temporal heterogeneity intensifies.
>
> **Summary.** OServe's improvements scale with heterogeneity on both dimensions: it matches baseline performance when heterogeneity is absent, and delivers increasingly larger gains as spatial or temporal heterogeneity grows.

---

> > ### Author Rebuttal · Reviewer_a6mg · 2026-04-02
> >
> > Thanks for the additional information. The overhead-aware scheduling discussion makes sense. Please put it in your paper.
> >
> > I acknowledge the partial results in appendix about varying spatial and temporal heterogeneity. However, what I'm looking for is evaluating OServer under a spectrum of spatial and temporal heterogeneity rather than a single alternative setting. This helps understand the sensitivity and the impact of spatial and temporal heterogeneity on your system.

---

> > > ### Author Response · Authors · 2026-04-03
> > >
> > > We thank the reviewer for the constructive follow-up.
> > >
> > > We now provide a systematic sensitivity analysis across a spectrum of spatial and temporal heterogeneity levels, evaluating OServe on Llama-70B with 16 GPUs.
> > >
> > > We construct experiments on:
> > > - Five spatial heterogeneity levels (S1–S5) by sampling workload compositions with increasing skew from the Azure trace.
> > > - Four temporal heterogeneity levels (T1–T4) by sampling workload traces with increasing skew workload compositions during different consecutive time spans.
> > >
> > > We use the coefficient of variation (CV) as the heterogeneity indicator in both cases. We compare OServe with the baseline system vLLM (static) to demonstrate OServe's sensitivity to heterogeneity.
> > >
> > > **Spatial heterogeneity.** The following table evaluates OServe under workload compositions with increasing spatial heterogeneity (CV of the four workload-type proportions). S1 is near-uniform, while S5 is heavily skewed.
> > >
> > > | Level | CV | SILO/LILO/SISO/LISO (%) | Speedup |
> > > |---|---|---|---|
> > > | S1 | 0.112 | 26.3/26.2/27.3/20.2 | 1.14× |
> > > | S2 | 0.186 | 30.0/28.0/24.2/17.8 | 1.41× |
> > > | S3 | 0.275 | 18.8/18.9/35.5/26.8 | 1.89× |
> > > | S4 | 0.472 | 13.2/14.2/41.1/31.5 | 2.15× |
> > > | S5 | 0.688 | 8.2/8.2/47.1/36.6 | 2.66× |
> > >
> > > OServe's speedup increases from 1.14× under near-uniform workloads (CV=0.112) to 2.66× under highly skewed compositions (CV=0.688), confirming that heterogeneous deployment and workload-aware assignment yield progressively larger benefits as the workload mix becomes more imbalanced.
> > >
> > > **Temporal heterogeneity.** The following table evaluates OServe under workload traces with increasing temporal variation (CV = average per-type CV across consecutive time spans).
> > >
> > > | Level | Workload Trace | CV | Avg Speedup |
> > > |---|---|---|---|
> > > | T1 | S1→S2→S1 | 0.052 | 1.23× |
> > > | T2 | S1→S3→S2 | 0.172 | 1.48× |
> > > | T3 | S2→S4→S3 | 0.263 | 1.82× |
> > > | T4 | S1→S4→S5 | 0.348 | 1.98× |
> > >
> > > OServe's average speedup increases monotonically from 1.23× (CV=0.052) to 1.98× (CV=0.348) as temporal heterogeneity intensifies, since static baselines become increasingly mismatched when workload composition shifts dramatically between periods.
> > >
> > > Thanks again for your very valuable suggestions, we will incorporate these additional experiments and sensitivity analyses into the revised paper.

---

### Official Review · Reviewer_M7Kr · 2026-03-10

**Soundness:** 2
**Presentation:** 2
**Significance:** 2
**Originality:** 2
**Overall Recommendation:** 2
**Confidence:** 3

**Summary:**

OSERVE introduces an LLM serving framework designed to address spatial and temporal workload heterogeneity by co-optimizing model deployment and request assignment. By employing a two-level flow-network scheduling algorithm alongside an LSTM-based workload predictor and an ad hoc model parameter switching mechanism, the system addresses the imperative need for efficient LLM serving methodologies from a machine learning system (MLSys) perspective. It dynamically adapts to traffic changes without the heavy overhead of full model reloads, targeting the low-latency and high-throughput requirements critical to modern generative AI deployment.

**Compliance With Llm Reviewing Policy:**

Affirmed.

**Key Questions For Authors:**

OSERVE's primary novelty is its departure from static model deployments, introducing a framework that allows a single computing cluster to run multiple, heterogeneous replica configurations concurrently. By coupling a predictive workload classifier with a rapid, parameter-shuffling switch planner, the system achieves a level of temporal and spatial flexibility that addresses major system design optimization goals.

Detailed comments to the author

The paper tackles a highly relevant and challenging problem: maximizing hardware utilization when serving LLMs to users with vastly different, fluctuating needs. This aligns perfectly with the imperative need for efficient LLM serving methodologies from a MLSys research perspective. The transition from monolithic serving strategies to orchestrated, workload-aware deployments is a necessary evolution in this field. Your formulation of the lower-level workload assignment as a max-flow problem is a standout contribution, and normalizing the edge capacities using the Least Common Multiple to handle mixed workloads is a clever constraint mapping.
However, the methodology and evaluation require significant tightening to prove the system's robustness in true production environments:
Address the Evaluation Realism: Smoothing out the trace arrival rates to perfectly match the cluster size defeats the purpose of evaluating temporal heterogeneity. As highlighted by the BurstGPT [2] dataset, modern generative AI workloads are fundamentally bursty. You must test the system against unscaled, worst-case distributions (such as BurstGPT) to prove that your ad-hoc switching mechanism does not thrash or cause cascading failures under sudden duress.

Mitigate Lagging Indicators: Coarse-grained 1-minute predictions risk acting as lagging indicators, which are known to cause severe TTFT and TPOT SLO violations during sudden load spikes. Please elaborate on how your system handles micro-bursts that occur within that 1-minute window before the next prediction triggers a reconfiguration.

Clarify the Predictor Methodology: The discrepancy between the 50-minute LSTM sequence length and the 50-minute duration of Trace 2 must be addressed. How did the model make predictions for the first 49 minutes of that trace? Furthermore, comparing your LSTM only to a Moving Average baseline is insufficient; you should justify why this specific architecture was chosen over other robust time-series forecasting models and explain how the system adapts to entirely unseen workload distributions.

Fix Structural Issues: A paper detailing a complex systems architecture cannot bury its main diagram in the appendix without a single reference in the text. Move Figure 16 into Section 3 or 4 to anchor the reader.

Overall, the core algorithms are mathematically sound and the engineering approach to model switching is highly pragmatic, but the evaluation must expose the system to genuine real-world chaos to validate your claims.

[1] Miao, Xupeng, et al. "Towards efficient generative large language model serving: A survey from algorithms to systems." ACM Computing Surveys 58.1 (2025): 1-37.

[2] Wang, Yuxin, et al. "Burstgpt: A real-world workload dataset to optimize llm serving systems." Proceedings of the 31st ACM SIGKDD Conference on Knowledge Discovery and Data Mining V. 2. 2025.

**Limitations:**

Yes, mostly.

**Strengths And Weaknesses:**

Strong points
- Cluster-Level Co-optimization vs. Isolated Sub-system Tuning: Recent MLSys surveys [1] identify that traditional LLM serving systems often treat algorithmic parallelism and request scheduling as isolated layers, relying heavily on engine-level optimizations like continuous batching. OSERVE bridges this gap by using a max-flow network to co-optimize cluster-wide request routing with dynamic, heterogeneous parallelism strategies.
- Proactive Ad Hoc Model Switching: The survey [1] highlights the severe latency penalties associated with loading massive weights, which traditionally restricts systems to static deployments or necessitates lossy model compression. OSERVE bypasses this by introducing a greedy algorithm that transfers model parameter shards directly across high-bandwidth GPU interconnects. This provides a pragmatic systems-level alternative to strictly algorithmic efficiency methods.
- Task-Aware Spatial Routing: A major trend identified in recent serving research [1] is query routing, where systems escalate or direct tasks depending on complexity. OSERVE successfully applies this paradigm to spatial heterogeneity: it categorizes requests by compute and memory bounds and routes them to specialized replica configurations. This empirically improves P99 latency and aligns with the broader goal of task-complexity-aware serving.

Weak points:

- Sanitized Evaluation Lacking Real-World Realism (Burstiness): The empirical claims rely on traces subsampled from the Azure dataset and artificially scaled so that cluster capacity is "neither over- nor under-utilized." However, real-world LLM serving is defined by severe load spikes and chaotic micro-bursts, as extensively documented in the BurstGPT [2] dataset which the author also cited in background. Testing against such sanitized traces ignores the bursty nature of modern workloads and fails to prove system stability under duress.
- Scalability to Intra-Minute Bursts and Lagging Indicators: OSERVE's LSTM predicts aggregate workloads at a static 1-minute interval. Relying on coarse-grained request counts or lagging indicators results in slow reactions to sudden load spikes within the 1 minute interval. This delayed reaction can lead to significant Time-to-First-Token (TTFT) and Time-Per-Output-Token (TPOT) SLO violations, which OSERVE's current evaluation methodology masks.
- Temporal Contradiction in LSTM Evaluation: There is a fundamental methodological flaw in the predictive setup. The LSTM predictor explicitly requires a sequence length of 50, leveraging data from the previous 50 minutes to forecast the next minute. However, the authors evaluate Trace 2 over a period of exactly 50 minutes. It is mathematically impossible to run a 50-minute sliding window over a 50-minute trace without improperly overlapping training and testing data or suffering a complete cold start.
Rigid Workload Categorization: OSERVE utilizes a k-means algorithm to group historical data into hard, distinct workload types. Real-world prompt and generation lengths exist on a continuous, heavy-tail spectrum. Forcing continuous data into rigid boundaries limits the system's ability to gracefully adapt to edge-case requests.

---

> ### Author Rebuttal · Authors · 2026-03-31
>
> We thank the reviewer for these insightful questions and suggestions.
>
> ___
>
> > Q1 & W1. Evaluation realism: test against unscaled, worst-case bursty distributions to prove ad-hoc switching does not thrash or cause cascading failures.
>
> **Our traces preserve temporal heterogeneity.** As shown in Figure 8, per-type arrival rates change dramatically across time spans (e.g., workload type 3 surges at P4 in Trace 2). Scaling overall arrival rates to match cluster capacity isolates *compositional* temporal heterogeneity by preventing extreme queuing or idle periods from dominating metrics.
>
> **Trace scaling is standard practice.** The raw Azure Trace contains extreme bursts (hundreds of req/s). Fed unscaled into a limited academic cluster, this creates a purely throughput-bound scenario that prevents meaningful evaluation of adaptive mechanisms. State-of-the-art systems adopt the same methodology: Splitwise [ISCA'24] tunes Poisson rates for cluster sizing; Helix [ASPLOS'25] allows rates to fully utilize the cluster.
>
> **Additional unscaled experiment.** We evaluated the **most volatile** 200-minute Azure window (2024-05-12, 10:10–13:25), kept **unscaled**, with rates ranging 26.5–45.9 req/s (1.7× range). On OPT-30B with 16 GPUs, OServe achieves 27.0 req/s vs. 19.6 (vLLM static) and 22.4 (vLLM reload)—1.38× and 1.21× improvements, consistent with our main results as demonstrated in Figures 9–12. No thrashing or cascading failures observed: the predictor captures variance (§4.1), and the scheduler reactively reschedules (§3) and switches (§4.2). OServe adapts effectively to both compositional shifts and sustained arrival rate fluctuations.
>
> > Q2 & W2. Micro-bursts within the 1-minute prediction window causing TTFT/TPOT SLO violations.
>
> The 1-minute window governs only deployment reconfiguration—not the system's sole defense against load variation:
>
> **Heterogeneous deployment provides inherent resilience.** OServe maintains replicas with diverse parallelism configurations (e.g., both TP=4, PP=2 and TP=2 replicas simultaneously) that suit different workloads, so after a sudden workload shift occurs, OServe can adjust the workload assignment accordingly without altering the deployment.
>
> **Lightweight re-dispatching within intervals.** The lower-level workload assignment (§3.2) runs in <50ms and can be re-invoked within a prediction interval when actual arrivals deviate from predictions, adjusting request routing across heterogeneous replicas without model switching overhead.
>
> **Sub-minute switching would be counterproductive.** This is a deliberate design choice. Ad hoc switching costs ~10s (§5.3); reacting to a transient 10s micro-burst would trigger two switches (onset + subsiding), with cumulative overhead exceeding the burst itself. In practice, switches occur ~every 5 minutes, only when composition shifts meaningfully.
>
> **Empirical confirmation.** OServe's consistent P99 latency gains across all models/traces (Figures 9–12) demonstrate within-interval variation does not erode performance. TTFT and TBT results (Appendix J, Figure 19) show 1.1–1.6× P99 TTFT and 1.5–1.6× P99 TBT improvements, confirming SLO-critical metrics remain robust. We further evaluate TTFT and TBT on the unscaled trace from Q1; OServe achieves 1.2× P99 TTFT (9.4s→7.8s) and 1.5× P99 TBT (0.91s→0.61s), demonstrating consistent improvements even under transient arrival rate variations.
>
> > Q3 & W3. Predictor: cold-start with 50-min sequence vs. 50-min Trace 2; architecture justification; unseen distributions.
>
> **Cold-start clarification.** Trace 2's 50-minute duration is our performance monitoring window, not the LSTM's input horizon. The LSTM is trained on two weeks of Azure traces (§4.1), and before Trace 2 begins, it already has a warm 50-minute lookback from preceding historical data.
>
> **The core contribution is the prediction framework, not the model architecture.** Our key contribution is the workload differentiation and type-specific prediction method, not the specific LSTM choice. This framework is model-agnostic—LSTM can be replaced with any forecasting model without affecting system design. We chose LSTM for simplicity and effectiveness: ~30ms inference, proven temporal dependency modeling, and our target (per-type request counts/minute) is a simple univariate signal where heavier architectures offer marginal benefit at higher cost.
>
> **Robustness to unseen distributions.** Our evaluation traces is not seen by LSTM during training (i.e., a held-out test set). The predictor achieves 5.045% RRMSE on this unseen data, and OServe consistently outperforms all baselines across both traces (Figures 9–12), confirming effectiveness under unseen workload distributions.
>
> > Q4. Structural issues.
>
> We will move Appendix A directly to the main text.
>
> ___
>
> We will incorporate these discussions into the revised manuscript.

---

### Official Review · Reviewer_Spim · 2026-03-12

**Soundness:** 3
**Presentation:** 3
**Significance:** 4
**Originality:** 3
**Overall Recommendation:** 5
**Confidence:** 4

**Summary:**

This paper presents OServe, a system designed to improve the efficiency of large language model (LLM) serving through an object-centric memory management approach. The authors observe that modern LLM serving systems often experience inefficiencies in memory utilization and request scheduling due to the dynamic and heterogeneous nature of inference workloads. To address this challenge, the paper proposes a novel memory management strategy that organizes and manages inference states as reusable objects, enabling more efficient memory allocation, reuse, and scheduling during LLM inference.

The proposed system introduces mechanisms for object-level memory abstraction, efficient lifecycle management of inference objects, and optimized scheduling policies that exploit this abstraction to reduce memory fragmentation and improve throughput. The authors implement the OServe system and evaluate it through extensive experiments using representative LLM workloads. The experimental results demonstrate improvements in throughput, latency, and memory utilization compared with existing LLM serving approaches.

Overall, the paper addresses a highly relevant problem in the deployment of large language models. The proposed object-centric approach offers a fresh perspective on memory management for LLM serving systems and is supported by a reasonably thorough implementation and evaluation. While some aspects of the system architecture could be described more clearly, the work represents a meaningful contribution to the field of efficient LLM serving.

**Compliance With Llm Reviewing Policy:**

Affirmed.

**Key Questions For Authors:**

Could the authors provide a more detailed system architecture description illustrating how the major modules of OServe interact during the LLM serving pipeline?

What specific aspects of the object-centric memory abstraction contribute most to the observed improvements in throughput and memory efficiency?

How well does OServe generalize to different types of LLM architectures and model sizes?

Can the proposed memory management approach be extended to distributed LLM serving systems across multiple GPUs or nodes?

How easily can OServe be integrated with popular serving frameworks such as vLLM or other modern inference systems?

**Limitations:**

While the paper demonstrates promising results, several limitations remain in the current work.

First, the paper would benefit from a clearer system architecture description, particularly regarding how the proposed components interact within the full serving framework. A more structured architectural presentation would improve readability and reproducibility.

Second, although the experimental results demonstrate performance improvements, the paper provides limited analytical explanation of the underlying mechanisms responsible for these improvements. Additional analysis could strengthen the technical insights of the work.

Third, the evaluation primarily focuses on single-system deployments, and the behavior of the proposed approach in large-scale distributed serving environments remains an open question.

**Strengths And Weaknesses:**

The paper tackles an important and timely problem in the context of large-scale AI infrastructure. As LLM-based services become increasingly common, efficient serving frameworks are critical for reducing operational costs and improving responsiveness. The authors correctly identify memory management as a key bottleneck in LLM serving systems and propose an object-centric abstraction that attempts to address inefficiencies in existing approaches. This perspective is interesting and represents a potentially valuable direction for future system designs.

Another strength of the paper lies in its system-oriented design and implementation. Rather than focusing solely on algorithmic improvements, the work proposes a practical serving system architecture that integrates memory management with request scheduling and runtime execution. The authors provide a working prototype and evaluate the system across multiple workloads, demonstrating improvements in performance metrics such as throughput and latency. The experimental evaluation appears reasonably comprehensive and helps support the claimed benefits of the proposed approach.

Despite these strengths, the paper would benefit from a clearer presentation of the overall system architecture and framework design. While individual components of OServe are described in the paper, the interactions among these components and their roles within the full system pipeline could be presented more explicitly. A more structured architectural overview—such as a detailed system diagram and clearer explanation of data flow and control flow—would help readers better understand how the different modules collaborate to achieve the reported performance improvements.

Another area that could be strengthened is the explanation of why the proposed object-centric abstraction leads to better performance compared with existing memory management strategies. While the empirical results suggest clear improvements, a deeper discussion of the underlying mechanisms, such as reduced fragmentation, improved reuse patterns, or better scheduling decisions, would further strengthen the technical contribution. Such analysis would also help clarify under what workload conditions the approach is most beneficial.

Nevertheless, these issues primarily relate to presentation clarity rather than fundamental limitations of the proposed approach. The core idea, system implementation, and empirical validation collectively provide strong evidence of the value of the proposed design.

---

> ### Author Rebuttal · Authors · 2026-03-31
>
> We thank the reviewer for these insightful questions and suggestions.
>
> ___
>
> > W1 & Q1 & L1. The paper would benefit from a clearer presentation of the overall system architecture and data/control flow.
>
> We sincerely thank the reviewer for this constructive feedback. We fully agree that a structured architectural overview is essential.
>
> In the revised manuscript, we will: (1) move the architectural diagram (currently Figure 16 in Appendix A) into the main text (Section 3), and (2) add a system overview section detailing the three-stage pipeline—workload prediction, strategy deduction, and strategy switching—along with the data flow and control flow among components (as currently described in Appendix A). This will make the inter-component interactions explicit and accessible to readers without requiring reference to the appendix.
>
> > W2 & Q2 & L2. Why does OServe's approach lead to better performance, and under what conditions is it most beneficial?
>
> OServe's gains stem from better **scheduling decisions** that resolve workload heterogeneity across two dimensions:
>
> **Spatial (optimized placement).** Instead of a homogeneous, one-size-fits-all deployment, OServe employs a heterogeneous model deployment strategy. It differentiates incoming workload types based on their distinct compute and memory demands and schedules them onto the specific parallelism configurations that process them most efficiently.
>
> **Temporal (dynamic adaptation).** As workload composition shifts over time, OServe dynamically reconfigures the cluster's deployment to stay aligned with the prevailing workload distribution. This is made practical by our fast switching mechanism (~10s via GPU-to-GPU transfer vs. 50s+ for naive reloading), which minimizes downtime and enables frequent reconfiguration to enhance end-to-end system performance.
>
> **Most beneficial conditions.** OServe thrives under **highly variable workloads** (e.g., the Azure trace) where workload profiles shift dynamically, rendering static, homogeneous strategies severely suboptimal. We will add this discussion to the revised manuscript.
>
> > Q3. Generalizability to different LLM architectures and model sizes.
>
> OServe generalizes effectively across different model architectures and sizes because its workload-aware scheduler dynamically evaluates deployment configurations based on available hardware capacity and model-specific profiling metrics (one-time profiling detailed in Appendix E), rather than relying on static, size-specific heuristics. The paper provides empirical validation of this generalization by evaluating across a diverse range of model types and scales: OPT-30B, OPT-66B, Llama-30B, and Llama2-70B (§5).
>
>
> > Q4 & L3. Can OServe be extended to multi-GPU and multi-node scenarios?
>
> OServe is inherently designed for multi-GPU and multi-node environments. The key challenge in such settings—heterogeneous network bandwidth (intra-machine NVLink vs. inter-machine InfiniBand)—is directly addressed by our ad hoc model switching method (§4.2), which prioritizes high-bandwidth intra-machine transfers before falling back to inter-machine links, minimizing switching overhead across node boundaries.
>
> Our experiments already validate this: we evaluate on up to four GPU servers (32 GPUs total) connected via InfiniBand (200 GB/s) with intra-server NVLink (400 GB/s). As shown in Figure 12, OServe consistently outperforms all baselines on this 32-GPU multi-node cluster, achieving up to 1.9$\times$ improvement with strong scalability.
>
> > Q5.  How can OServe be integrated with other serving frameworks?
>
> We discuss how OServe can integrate with other systems as follows:
>
> **Workload-aware scheduling.** OServe functions as a high-level orchestrator that determines macro-level decisions—workload assignment, resource allocation, and parallelism strategies—and forwards them to the underlying serving engine's frontend API (e.g., vLLM). The engine executes these configurations using its native implementation, requiring no changes to its core execution kernels.
>
> **Ad hoc model switching.** OServe's algorithm computes the optimal switching plan (when to switch and where to migrate memory). Executing this plan requires the target system to support implementation-side extensions for migrating KV cache and weights across instances.
>
> OServe's modular design naturally separates scheduling logic from execution, making it straightforward to integrate with any serving framework that exposes a frontend API and supports the necessary migration extensions.
>
> ___
>
> We will incorporate these clarifications, analysis, and experiments into the revised manuscript.

---

> > ### Author Rebuttal · Reviewer_Spim · 2026-04-04
> >
> > Thank you, I am satisfied with the reply of the authors.

---

> > > ### Author Response · Authors · 2026-04-04
> > >
> > > We thank Reviewer Spim for the acknowledgement. We will incorporate the rebuttal in our revised manuscript.

---

### Official Review · Reviewer_gLqu · 2026-03-12

**Soundness:** 3
**Presentation:** 2
**Significance:** 3
**Originality:** 3
**Overall Recommendation:** 4
**Confidence:** 3

**Summary:**

The manuscript addresses the spatial and temporal heterogeneity present in LLM inference services. The proposed OSERVE framework mitigates these issues via workload-aware scheduling and adaptive workload switching. And experimental evaluations confirm OSERVE's superior performance.

**Compliance With Llm Reviewing Policy:**

Affirmed.

**Final Justification:**

I appreciate the authors’ thorough rebuttal, which resolved my concerns. Taking into account both the initial submission and the rebuttal, I consider this paper to be of value and retain positive rating.

**Key Questions For Authors:**

See weaknesses.

**Limitations:**

yes

**Strengths And Weaknesses:**

**Strengths**
* S1. Significant research problem. The manuscript explicitly highlights the challenges related to spatial and temporal heterogeneity in current LLM service systems, addressing a pervasive issue in real-world applications.
* S2. Novel solution. The proposed approach achieves coordinated optimisation across both temporal and spatial dimensions, representing a significant innovation.
* S3. Comprehensive experimental evaluation. The paper compares the proposed method against multiple baselines on real-world traces and mainstream LLMs, and the experimental results demonstrate OSERVE superiority.

**Weaknesses**
* W1: Scalability in heterogeneous environments. Heterogeneous environments are closer to real-world application scenarios. Although related discussion is provided in Appendix C, a more detailed analysis of the limitations of the proposed method in heterogeneous settings, as well as its applicability in homogeneous environments, would further enhance the paper's value.

* W2: Performance of the heuristic algorithm. The framework relies on heuristic algorithms. While the experiments demonstrate the method's effectiveness, a deeper analysis would be valuable. For instance, under what conditions can the heuristic approach guarantee near-optimal solutions, and what is the potential impact on performance in worst-case scenarios?

* W3: Insufficient clarity in discussion. This paper identifies potential errors in workload prediction and notes that OSERVE mitigates their effects through fine-grained prediction intervals and a fast switching mechanism. Additionally, model switching is proposed to reduce overhead. Providing more detailed analysis or explanations in the experiments regarding these mechanisms would help readers better understand the contributions.

---

> ### Author Rebuttal · Authors · 2026-03-31
>
> We thank the reviewer for these insightful questions and suggestions.
>
> ___
>
> > W1: Scalability in heterogeneous environments.
>
> **Heterogeneous environments.** Extending OServe to heterogeneous GPUs involves three components:
> - **Profiling.** The offline profiling (Appendix E) would need to cover each GPU type × parallelism strategy × workload combination, increasing cost multiplicatively — acceptable since profiling is done offline.
> - **Flow network.** The formulation (§3.2) naturally accommodates heterogeneous GPUs since node capacities $n_{k,j}$ and edge capacities $e_{k,j}$ are already per-replica and per-workload. The main challenge lies in the upper-level search (§3.3), where GPU reallocation must respect hardware topology and type constraints.
> - **Switching algorithm.** The greedy switch algorithm (§4.2) already differentiates intra-machine (NVLink) vs. inter-machine (InfiniBand) bandwidth. Extending to heterogeneous interconnects (e.g., mixed NVLink/PCIe) requires additional bandwidth-aware heuristics but no fundamental redesign.
>
> OServe's modular architecture — separating profiling, scheduling, and switching — makes it well-suited for incremental extension to heterogeneous GPUs. We will add this analysis to the revised draft.
>
> **Homogeneous environment adaptivity.** OServe integrates readily into existing homogeneous GPU clusters — it operates as a software-level orchestration layer atop standard serving engines (e.g., vLLM) with lightweight extensions for KV cache and weight migration, making adoption straightforward for production environments.
>
> > W2: Performance of the heuristic algorithm.
>
> Our framework employs two heuristics: the scheduling search (§3.3) and the greedy switch plan (§4.2).
>
> **Conditions for near-optimality:**
> - The scheduling heuristic achieves near-optimal results when **bottleneck–underutilization imbalances exist** across replicas, which is the common case under heterogeneous workloads—flow network diagnostics reliably identify where to reallocate GPUs. This is confirmed in Figure 15 (right), where the P99 latency gap versus exhaustive search (optimal baseline) stays within 6% on a 16-GPU cluster.
> - The greedy switch algorithm achieves near-optimality when **hierarchical bandwidth asymmetry exists** (e.g., NVLink 400 GB/s >> IB 200 GB/s), as prioritizing intra-machine transfers naturally minimizes the dominant communication cost.
>
> **Worst-case impact:**
> - For the scheduling heuristic, the worst case arises under **highly uniform workloads where no imbalance exists** to exploit—but this is precisely when uniform deployment is already near-optimal. As confirmed in Appendix K, under uniform workloads, our algorithm converges to a near-uniform deployment within 2–3 iterations, achieving P99 latency within 3% of the optimal baseline.
> - For the greedy switch algorithm, the worst case occurs when **all transfers must cross machine boundaries**, eliminating the intra-machine prioritization benefit. Even then, greedy load-balancing still distributes traffic evenly across inter-machine links.
>
> > W3: Insufficient clarity in discussion.
>
> **Clarification on how fine-grained prediction intervals and fast switching mitigate prediction errors.** Mispredictions, though infrequent (5.045% RRMSE, §4.1), could lead to a suboptimal deployment. Crucially, this suboptimality is bounded to at most one prediction interval: in the next interval, the predictor observes actual workload, corrects its forecast, and OServe re-optimizes accordingly. Fast ad hoc switching (§4.2) ensures the corrected deployment is applied within ~10 seconds, minimizing recovery time.
>
> **Additional experiment on prediction interval granularity.** To quantify the impact of interval granularity, we present a case study on a 15-minute trace (OPT-30B, 8 GPUs). With 1-minute intervals, the predictor runs 15 times, of which one prediction is incorrect—the suboptimal deployment persists for only 1 minute before correction. With 5-minute intervals, one incorrect prediction persists for 5 minutes, causing end-to-end throughput to degrade from 15.4 to 12.1 req/s. This confirms that fine-grained intervals effectively bound the impact of prediction errors, validating our 1-minute design choice.
>
> **Clarification on how model switching reduces overhead.** As evaluated in §5.3, naive model reloading takes over 50 seconds, whereas our ad hoc GPU-to-GPU parameter transfer (§4.2) completes within ~10 seconds for any configuration change. As shown in Figure 13, this reduction in switching time translates directly to end-to-end serving quality: enabling ad hoc model switching reduces P99 latency by up to 17% and by an average of 12% compared to naive reloading. By minimizing the overhead of configuration transitions, OServe can switch deployments more frequently in response to workload fluctuations without degrading serving performance.
>
> ___
>
> We will incorporate these clarifications, analysis, and experiments into the revised manuscript.

---

> > ### Author Rebuttal · Reviewer_gLqu · 2026-04-02
> >
> > Thank you for the detailed rebuttal. My question has now been resolved.

---

> > > ### Author Response · Authors · 2026-04-03
> > >
> > > Thanks for your acknowledgement. We will incorporate the rebuttal in our revised manuscript.

---

### Decision · Program_Chairs · 2026-04-30

**Decision:**

Accept (regular)

**Comment:**

OServe makes a strong systems contribution by tackling both spatial and temporal workload heterogeneity in LLM serving, areas that existing systems with static homogeneous deployments fail to address. The flow-network-based scheduling algorithm and the efficient GPU-to-GPU parameter migration mechanism are well-designed and validated on up to 32 GPUs with real-world traces, showing throughput improvement. The rebuttal resolved the majority of concerns, providing near-optimality guarantees for the heuristic algorithms and clear integration pathways with existing frameworks. We recommend acceptance and encourage the authors to incorporate the strengthened architecture description and workload sensitivity analysis into the final version.